# Memory Efficient Transformer Adapter for Dense Predictions

**Dong Zhang**[1,2]**, Rui Yan**[3]**, Pingcheng Dong**[1]**, Kwang-Ting Cheng**[1]
[1]The Hong Kong University of Science and Technology
[2]AI Chip Center for Emerging Smart Systems (ACCESS), [3]Nanjing University
`{dongz,timcheng}@ust.hk;ruiyan@nju.edu.cn;pingcheng.dong@connect.ust.hk`

## Abstract

While current Vision Transformer (ViT) adapter methods have shown promising accuracy, their inference speed is implicitly hindered by inefficient memory access operations, *e.g.*, standard normalization and frequent reshaping. In this work, we propose META, a simple and fast ViT adapter that can improve the model's memory efficiency and decrease memory time consumption by reducing the inefficient memory access operations. Our method features a memory-efficient adapter block that enables the common sharing of layer normalization between the self-attention and feed-forward network layers, thereby reducing the model's reliance on *normalization operations*. Within the proposed block, the cross-shaped self-attention is employed to reduce the model's frequent *reshaping operations*. Moreover, we augment the adapter block with a lightweight convolutional branch that can enhance local inductive biases, particularly beneficial for the dense prediction tasks, *e.g.*, object detection, instance segmentation, and semantic segmentation. The adapter block is finally formulated in a cascaded manner to compute diverse head features, thereby enriching the variety of feature representations. Empirically, extensive evaluations on multiple representative datasets validate that META substantially enhances the predicted quality, while achieving a new state-of-the-art accuracy-efficiency trade-off. Theoretically, we demonstrate that META exhibits superior generalization capability and stronger adaptability.

## 1 Introduction

State-of-the-art computer vision models typically follow the intuitive paradigm of pre-training on large single-modal general datasets, followed by fine-tuning on local task-specific datasets to achieve promising accuracy (Chen et al., 2021; Zhang et al., 2024; Radford et al., 2021; He et al., 2022; Zhang et al., 2023). However, this defacto paradigm requires downstream models to load the entire pre-trained model for fine-tuning, which may result in several undesirable drawbacks such as poor structural flexibility, large optimization gaps, and limited applicability in certain scenarios (Jie & Deng, 2023; Du et al., 2022; Jie & Deng, 2022). More importantly, these drawbacks have become more prominent and urgent problems that need to be addressed for dense prediction tasks, particularly in the current context of significant model size growth (Gao et al., 2023; Zhang et al., 2022b; Radford et al., 2021; Zhang et al., 2023; Kirillov et al., 2023).

Recently, with the increasing dominance of Vision Transformer (ViT) architectures (Han et al., 2022; Liu et al., 2021; Khan et al., 2022), following the parameter-efficient transfer learning mechanism (Houlsby et al., 2019), ViT adapter has become a central approach to learning vision-specific inductive biases from pre-trained ViT models (Hu et al., 2022; Jie & Deng, 2023; Chen et al., 2022b; Ma et al., 2024; Luo et al., 2023; Shao et al., 2024), successfully addressing the drawbacks associated with the pre-training followed by fine-tuning paradigm. The progressive ViT adapter enables downstream ViT models to achieve promising accuracy levels that are comparable to, or even higher than, those achieved by fine-tuning the entire model. In particular, thanks to the utilization of plain ViT models, which contain unique multi-modal information, adapters based on these models can effectively promote downstream models to learn beneficial semantic-rich feature representations (Touvron et al., 2021; Hu et al., 2022; Kirillov et al., 2023; Dosovitskiy et al., 2020). For example, ViT adapter under the plain ViT models has been successfully applied in multiple computer vision tasks,

*e.g.*, object detection (ODet) (Li et al., 2022b), instance segmentation (ISeg) (Liu et al., 2024), and semantic segmentation (SSeg) (Xie et al., 2021).

Despite the significant progress made by existing ViT adapters (Chen et al., 2022b; Liu et al., 2024; Jie & Deng, 2023; Marouf et al., 2024), their inference speed is still somewhat unfavorable, which limits their implementations on edge computing devices (Dong et al., 2024a) and applications for real-time recognition scenarios (Marouf et al., 2024). Recently, it has been revealed that in addition to computation and parameter complexity (*e.g.*, FLOPs and #Params.), inefficient memory access operations (Marouf et al., 2024; Liu et al., 2023), such as standard normalization and frequent reshaping operations, play a critical role in hindering the ViT's inference speed (He & Hofmann, 2024; Pan et al., 2022; Fournier et al., 2023; Shi et al., 2023). In other words, the inefficient planning of memory access may cause delays and prevent models from fully utilizing the computing power of GPUs or CPUs, resulting in a significant negative impact on the speed of ViT models (Liu et al., 2023; Dao et al., 2022; Venkat et al., 2019; Gu et al., 2021; Ivanov et al., 2021). However, these inefficient memory access operations are usually overlooked factors during the network design process for dense prediction tasks. In this work, we explore how to solve this problem and accelerate the inference speed of ViT adapters in downstream models. *Our solution is to decrease memory time consumption by reducing layer normalization and frequent reshaping operations.*

We propose a simple and fast memory efficient transformer adapter (META) that can improve the model's memory efficiency and decrease memory time consumption by reducing the inefficient memory access operations. As illustrated in Figure 1, our META demonstrates advantages over existing ViT models and ViT adapters in terms of memory consumptions and inference time costs.

| | ViT | ViT Adapter | META(Ours) |
|---|---|---|---|
| **Training Params.** | ☒ High | ☑ Low | ☑ Low |
| **Application gaps** | ☒ High | ☑ Low | ☑ Low |
| **Memory consumptions** | ☒ High | ☒ High | ☑ Low |
| **Inference time costs** | ☒ High | ☒ High | ☑ Low |

Figure 1: Qualitative performance comparisons of different models with respect to training parameters, application gaps, memory access costs, and inference time costs.

The main contribution of this work is the proposal of a *memory-efficient adapter block* that shares normalization operations between the self-attention and feed-forward network layers, which exist in a parallel manner (*Ref.* Sec. 3.2). Within this block, the cross-shaped self-attention is employed to reduce the reliance for frequent reshaping operations. Consequently, the proposed two reverse designs are capable of significantly reducing the memory consumption of the ViT adapter, resulting in an improved inference speed. Moreover, to enrich local inductive biases for dense predictions, a *lightweight convolutional branch* is introduced into the *memory-efficient adapter block*. In the process of interacting with the ViT backbone, a *cascaded mechanism* is further proposed to compute different head features, which can enhance the diversity of the obtained feature representations. We conduct extensive experiments for ODet, ISeg and SSeg on two challenging datasets, namely MS-COCO (Lin et al., 2014) and ADE20K (Zhou et al., 2017), for evaluating our META. The obtained results demonstrate that META substantially enhances prediction quality, attains a new state-of-the-art accuracy level, reduces the number of parameters and memory consumption requirements, and achieves faster inference speeds. Theoretically, we also prove that META can exhibit superior generalization capability and stronger adaptability compared to existing ViT adapter methods[1].

## 2 RELATED WORK

**Vision Transformer (ViT).** Current ViT models for state-of-the-art computer vision tasks can be roughly divided into two camps: *plain ViT models*, which are designed to learn general vision features (*e.g.*, ViT (Dosovitskiy et al., 2020), DeiT (Touvron et al., 2021) and TDE Transformer (Touvron et al., 2021)), and *hierarchical ViT models*, which are designed to learn vision-specific features (*e.g.*, Swin Transformer (Liu et al., 2021) and PVT (Wang et al., 2021)). Both camps of ViT models have their pros and cons. For example, the *hierarchical ViT models* can learn powerful vision-specific feature representations, which makes them usually perform better than the *plain ViT models* on accuracy (Han et al., 2022; Khan et al., 2022; Kirillov et al., 2023). However, their unvarnished disadvantage is the lack of multi-modal pre-training information for dense predictions (Chen et al., 2022b). On the other hand, although the *plain ViT models* have weak prior assumptions, which results in lower accuracy compared to the *hierarchical ViT models*, their multi-modal pre-training in-

---

[1]Due to page limitations, the theoretical analysis will been provided in the supplementary materials.

formation has great potential to provide semantic-rich feature representations for downstream models (Guo et al., 2022; Zhang et al., 2023; He et al., 2022; Zhang et al., 2024; Jia et al., 2022). In this paper, we focus on the *plain ViT models* for dense predictions. Our contribution is the proposal of a straightforward and memory efficient ViT adapter, starting from the reduction of inefficient memory access operations.

**ViT for Dense Predictions.** Dense predictions, including ODet, ISeg, and SSeg, aim at predicting a semantic mask where each pixel/object in the given image is assigned a class label (Ranftl et al., 2021; Zhang et al., 2023; Vandenhende et al., 2021). For ISeg, the obtained mask also distinguishes different objects of the same class. With the development of ViT technologies, dense prediction models based on ViT have become one of the default choices for state-of-the-art methods (Liu et al., 2021; Wang et al., 2021; Dosovitskiy et al., 2020). These methods can be mainly divided into two categories: those based on Transformers and those that incorporate a combination of CNN layers. The former, *e.g.*, SegFormer (Strudel et al., 2021), HRViT (Gu et al., 2022), Segvit (Zhang et al., 2022a), Swin UNet (Hatamizadeh et al., 2021), and ISTR (Hu et al., 2021), extract features from a given image and connect with task-specific head networks to achieve specific recognition purposes. The latter compensates for the lack of local inductive bias in the pure ViT architecture by introducing local convolutional layers, and representative methods include ConFormer (Peng et al., 2021), CVT (Wu et al., 2021), NextViT (Li et al., 2022a), and CAE-GReaT (Zhang et al., 2023). While the former can capture long-range dependencies and achieve favorable recognition capability, they require sufficient training samples to optimize the model, and further improvement is needed on memory access and inference speed. To this end, we propose a solution to the problem of limited inference speed caused by frequent memory consumption using a pre-trained ViT model.

**ViT Adapters.** Adapter methods are originally derived from the NLP tasks, such as PALs (Stickland & Murray, 2019) and NLP adapter (Houlsby et al., 2019). These methods add a small number of trainable modules to a pre-trained network, so that downstream fine-tuning models can quickly adapt to specific datasets and tasks via a parameter-efficient transfer learning manner (Hu et al., 2022; Houlsby et al., 2019; Chen et al., 2022b; Shen et al., 2023). In the computer vision domain, ViT adapters have also adopted the similar paradigm (Jie & Deng, 2023; Chen et al., 2022b; Shao et al., 2024; Luo et al., 2023). ViT adapter method is used to fine-tune large-scale plain ViT models with a small number of trainable parameters and achieve promising accuracy (Jie & Deng, 2023; Li et al., 2022b; Chen et al., 2022b). Inspired by (He & Hofmann, 2024; Jie & Deng, 2022; Chen et al., 2022b; Liu et al., 2023; Mercea et al., 2024), we propose an efficient solution to enhance the memory efficiency and minimize memory time consumption of ViT adapters. We also provide theoretical analysis to demonstrate the superior generalization and adaptability of our method.

# 3 MEMORY EFFICIENT TRANSFORMER ADAPTER (META)

## 3.1 OVERALL ARCHITECTURE AND CONFIGURATIONS

As illustrated in Figure 2, the network takes an arbitrary RGB image $\mathbf{I} \in \mathbb{R}^{H \times W \times 3}$ as input and predicts a semantic mask $\mathbf{O} \in \mathbb{R}^{H \times W \times C}$ as output. $H$ and $W$ denotes the image height and width, respectively, and $C$ denotes the class size of the used dataset. Following (Jie & Deng, 2023; Li et al., 2022b; Chen et al., 2022b), the whole network mainly consists of two parts: the upper part is a pre-trained plain ViT model (*e.g.*, ViT (Li et al., 2021)), which consists of 4 blocks. The $i$-th ($i = 1, 2, 3, 4$) block contains a set of semantic features $\mathbf{F}_{vit}^{i}$ with the spatial size of $1/4$, $1/8$, $1/16$, and $1/32$ of $\mathbf{I}$, respectively. The lower part is a trainable ViT adapter, which includes a spatial prior module as in (Chen et al., 2022b) for $\mathbf{I}$'s encoding, where the encoded features within each block denote $\mathbf{F}_{sp}^{i}$, as well as a set of cascaded memory efficient adapter (MEA) injectors (*Ref.* Sec. 3.3) and cascaded MEA extractors (*Ref.* Sec. 3.4) that act on each ViT backbone block. The "cascaded" refers to a computational scheme that is incorporated into the different heads of the proposed injectors/extractors, which can enhance the diversity of the obtained feature representations. Concretely, for the $i$-th trainable MEA injector, it injects the updated task-specific features back into the $i$-th ViT block and into the $i$-th extractor (as shown in Figure 2 (b)). For the $i$-th trainable MEA extractor, it extracts the $i$-th general features for the $(i + 1)$-th injector (as shown in Figure 2 (c)). This process is repeated until the final ViT block is reached. Due to the incorporation of strategies aimed at reducing inefficient memory access operations in our method, which exhibits enhanced memory efficiency and reduced memory time consumption compared to existing ViT adapters, consequently

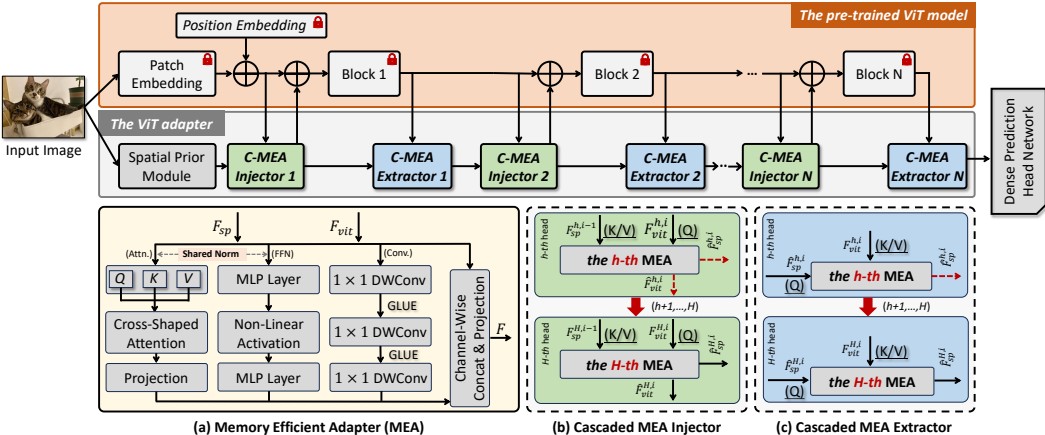

Figure 2: Overall architecture of META. Our primary contribution is the proposal of a MEA block in (a), which serves as the fundamental component for the injector in (b) and extractor in (c).

leading to accelerated inference speed. We will provide a detailed explanation of the basic component and mechanism of the MEA block in the following Sec. 3.2 and how it can be deployed in the MEA injectors and MEA extractors to form a cascaded scheme.

## 3.2 MEA BLOCK

The MEA block aims to interact the features extracted from the ViT backbone and the spatial prior module. Our block consists of the attention (*i.e.*, Attn) branch, the feed-forward network (*i.e.*, FFN) branch, and the lightweight convolutional (*i.e.*, Conv) branch. The three branches exist in a parallel form, which is beneficial for computing in GPUs. As shown in Figure 2 (a), consider an arbitrary block, the input is $\mathbf{F}_{sp}$ and $\mathbf{F}_{vit}$, and the output $\mathbf{F}$ is obtained by concatenating the features from the Attn, FFN, and Conv branches along the channel dimension, and then passing them through a feature projection layer. Therefore, this process can be formulated as:

$$\mathbf{F} = \mathrm{Conv}_{3\times3}(\mathrm{Concat}((\underbrace{\mathrm{A}(\mathbf{F}_{sp}, \mathbf{F}_{vit})}_{\textbf{\textcolor{red}{Attn Branch}}}; \underbrace{\mathrm{F}(\mathbf{F}_{sp}, \mathbf{F}_{vit})}_{\textbf{\textcolor{blue}{FFN Branch}}}; \underbrace{\mathrm{C}(\mathbf{F}_{sp}, \mathbf{F}_{vit})}_{\textbf{\textcolor{orange}{Conv Branch}}}; \mathbf{F}_{sp}; \mathbf{F}_{vit})), \tag{1}$$

where $\mathrm{A}(\cdot)$, $\mathrm{F}(\cdot)$, and $\mathrm{C}(\cdot)$ denote the operation of the Atte, FFN, and Conv branch, respectively. In MEA, these three branches exist in a parallel manner. $\mathrm{Concat}(\cdot)$ denotes the feature concatenation operation along the channel dimension, and $\mathrm{Conv}_{3\times3}(\cdot)$ denotes a $3 \times 3$ convolution with the output channel size of 256 for feature projection. In particular, to construct an memory efficient block, following (He & Hofmann, 2024; Wang & Komatsuzaki, 2021), *the Attn and FFN branches are subjected to a shared layer normalization operation, which leads to a decrease in memory time consumption associated with the normalization operations.*

**Attn Branch.** In our work, we adopt the cross-shaped self-attention (CSA) to reduce the model's access to the memory unit, which allows for *the computation of the attention matrix across different spatial dimensions without requiring the input tensor to be frequently reshaped* (Dong et al., 2022; Tu et al., 2022). Specifically, CSA first performs self-attention separately along the horizontal and vertical dimensions of the given features. The outputs $\mathrm{A}_H(\mathbf{F}_{sp}, \mathbf{F}_{vit})$ and $\mathrm{A}_V(\mathbf{F}_{sp}, \mathbf{F}_{vit})$ of these two parallel groups are then concatenated along the channel dimension, followed by a feature projection operation via a $3\times3$ convolution with the output channel size of 256, to form the output $\mathrm{A}(\mathbf{F}_{sp}, \mathbf{F}_{vit})$. For the horizontal self-attention, the input features $\mathbf{F}_{sp}$ and $\mathbf{F}_{vit}$ are first subjected to the shared layer normalization, and then divided into $M$ non-overlapping horizontal stripes, where each stripe has the spatial size of $s \times W$ (*i.e.*, $s = H/M$). Then, the self-attention (Vaswani et al., 2017) is performed on each stripe, which can be formulated as:

$$\mathrm{A}_H^m(\mathbf{F}_{sp}, \mathbf{F}_{vit}) = \mathrm{SA}(\mathrm{LN}(\mathbf{F}_{sp}^m W^Q), \mathrm{LN}(\mathbf{F}_{vit}^m W^K), \mathrm{LN}(\mathbf{F}_{vit}^m W^V)), \tag{2}$$

where $\mathrm{LN}(\cdot)$ denotes the shared layer normalization operation. $\mathrm{SA}(\cdot)$ denotes the classical single head self-attention (Vaswani et al., 2017). $m$ denotes the $m$-th stripe and $m = 1, 2, ..., M$. $W^Q$, $W^K$, and $W^V$ are used to project the input features into the queries, keys, and values spaces, respectively. The sub-attention results of different stripes are merged together to form the output

$A_H(\mathbf{F}_{sp}, \mathbf{F}_{vit})$ of the horizontal group. In equation 2, we choose $\mathbf{F}_{sp}$ as queries and $\mathbf{F}_{vit}$ as keys and values for the sake of example. In our injector and extractor in Sec 3.3 and Sec 3.4, the roles of $\mathbf{F}_{sp}$ and $\mathbf{F}_{vit}$ can be swapped depending on the specific requirements. The attention computation along the vertical dimension $A_V(\mathbf{F}_{sp}, \mathbf{F}_{vit})$ is also performed in a similar fashion.

**FFN Branch.** Following the common setting (Vaswani et al., 2017; Tolstikhin et al., 2021), to enable interaction between the obtained features in the channel dimension, our FFN branch consists of two MLP layers and a non-linear activation layer (Saxe et al., 2013; He & Hofmann, 2024), where the MLP layer consists of a sequential arrangement of two $3 \times 3$ convolutional layers.

The input of the FFN branch is $\mathbf{F}_{sp}$ and $\mathbf{F}_{vit}$, which are sequentially processed by feature concatenation along the channel dimension, a $3 \times 3$ convolution with the output channel size of 256, and the shared layer normalization. This process can be expressed as:

$$F(\mathbf{F}_{sp}, \mathbf{F}_{vit})_{\text{Tem}} = \text{LN}(\text{Conv}_{3\times3}(\text{Concat}(\mathbf{F}_{sp}; \mathbf{F}_{vit}))). \tag{3}$$

The temporary $F(\mathbf{F}_{sp}, \mathbf{F}_{vit})_{\text{Tem}}$ are then passed through a MLP layer, a non-linear activation layer (Saxe et al., 2013), and another MLP layer to obtain the output $F(\mathbf{F}_{sp}, \mathbf{F}_{vit})$. Hence, this can be expressed as:

$$F(\mathbf{F}_{sp}, \mathbf{F}_{vit}) = \text{MLP}(\text{NLA}(\text{MLP}(F(\mathbf{F}_{sp}, \mathbf{F}_{vit})_{\text{Tem}}))), \tag{4}$$

where $\text{MLP}(\cdot)$ denotes the MLP layer operation, and $\text{NLA}(\cdot)$ denotes the non-linear activation layer (Saxe et al., 2013; He & Hofmann, 2024).

**Conv Branch.** Empirical evidence has demonstrated that incorporating local inductive biases into ViT models can provide notable benefits for visual tasks involving dense prediction (Peng et al., 2021; Wu et al., 2021; Li et al., 2022a; Zhang et al., 2023). To achieve this goal, we introduce a lightweight convolutional branch into the ViT adapter. Specifically, the Conv branch also takes into $\mathbf{F}_{sp}$ and $\mathbf{F}_{vit}$ as the input, which consists of three $1 \times 1$ depth-wise convolutions concatenated in series, with a GLUE layer used to activate the features between every two convolutions (Saxe et al., 2013; He & Hofmann, 2024). The computation process of the Conv branch is formulated as:

$$C(\mathbf{F}_{sp}, \mathbf{F}_{vit}) = \text{DC}(\text{GLU}(\text{DC}(\text{GLU}(\text{DC}(\text{Concat}(\mathbf{F}_{sp}; \mathbf{F}_{vit})))))), \tag{5}$$

where $\text{DC}(\cdot)$ denotes the $1 \times 1$ depth-wise convolution with the output channel size of 256, and $\text{GLU}(\cdot)$ denotes the GLUE layer. With the help of shared layer normalization and CSA, MEA becomes a memory-efficient module with rich local inductive biases. In the following sections, we will provide detailed instructions on how to deploy our MEA block on injectors and extractors.

### 3.3 CASCADED MEA INJECTOR

As illustrated in Figure 2 (b), the cascaded MEA injector adopts the paradigm of self-attention in its computational process, where $\mathbf{F}_{vit}^{i}$ is used as the query, and $\mathbf{F}_{sp}^{i-1}$ generated by the last extractor is used as the key and value. Particularly, to enhance the diversity of the obtained attention maps, a cascaded mechanism is proposed to compute different head features in the cascaded MEA injector. Specifically, we first divide the given features into $H$ parts along the channel dimension in a multi-head manner, following the classic self-attention (Vaswani et al., 2017), where $H$ is set to 16 in our work. In the computation process of each head, the output of the $h$-th head $\hat{\mathbf{F}}_{sp}^{h,i}$ and $\hat{\mathbf{F}}_{vit}^{h,i}$ is added into the input features of the next $(h+1)$-th head $\mathbf{F}_{sp}^{h+1,i-1}$ and $\mathbf{F}_{vit}^{h+1,i}$ to be used in the calculation of subsequent self-attention features, where $h = 1, 2, ..., H$. The cascaded process continues until the feature from the last head is included in the computation. Finally, the features obtained from these heads are concatenated along the channel dimension and projected through a $3 \times 3$ convolution layer before being outputted as the output of the $i$-th cascaded MEA injector. Besides, since the MEA block has only one output, $\hat{\mathbf{F}}_{sp}^{h,i} = \hat{\mathbf{F}}_{vit}^{h,i}$ in our cascaded MEA injector.

### 3.4 CASCADED MEA EXTRACTOR

As illustrated in Figure 2 (c), following (Marouf et al., 2024; Chen et al., 2022a; Dong et al., 2024b), our $i$-th cascaded MEA extractor of the $h$-th head takes the $\hat{\mathbf{F}}_{sp}^{h,i}$ generated by the injector and $\mathbf{F}_{vit}^{h,i}$ as input, where $\hat{\mathbf{F}}_{sp}^{h,i}$ is used as the query, $\mathbf{F}_{vit}^{h,i}$ is used as the key and value. The output of the

$i$-th cascaded MEA extractor is $\mathbf{F}_{sp}^{h,i}$. Similarly, the cascaded mechanism is applied following the cascaded MEA injector until the feature from the last head is included in the computation.

# 4 EXPERIMENTS

## 4.1 DATASETS AND EVALUATION METRICS

**Datasets.** To facilitate a fair result comparison with existing methods, we conduct experiments, including the ablation analysis, on two commonly used datasets: MS-COCO (Caesar et al., 2018) for ODet and ISeg, and ADE20K (Zhou et al., 2017) for SSeg. Due to page limitations, the implementation details of these datasets will be given in the supplementary materials.

**Evaluation metrics.** The commonly adopted average precision (AP) and mean intersection-over-union (mIoU) are used to assess the model accuracy for ODet (AP$^b$)/ISeg (AP$^m$) and SSeg, respectively. Besides, to evaluate the efficiency, model Parameters (#P), floating point operations (FLOPs), memory consumption (MC) of the adapter, and frames per second (FPS) are also adopted. The reported inference results are measured by A100 GPUs with per-GPU batch size 2.

## 4.2 EXPERIMENTS ON OBJECT DETECTION (ODET) AND INSTANCE SEGMENTATION (ISEG)

**Baselines and settings.** As in (Chen et al., 2022b; Xiong et al., 2024; Jie & Deng, 2022; Marouf et al., 2024), Mask R-CNN (He et al., 2017), Cascade Mask R-CNN (Cai & Vasconcelos, 2019), ATSS (Zhang et al., 2020), and GFL (Li et al., 2020) are employed as the baseline models, where the pre-trained ViT (Li et al., 2022b) is used as the backbone. All the baseline models are pre-trained on ImageNet-1k by default (Deng et al., 2009). Unless otherwise specified, these baselines are set up to be consistent with their papers and the settings of the ViT-Adapter (Chen et al., 2022b) method.

**Comparisons with state-of-the-art (SOTA) methods.** Result comparisons with SOTA methods with Mask R-CNN (He et al., 2017) for ODet and ISeg are shown in Table 1. From this table, we can obtain the following observations and conclusions: *i*) Compared to the experimental results of the Mask R-CNN (He et al., 2017) model with ViT (Li et al., 2021), our proposed META can consistently improve accuracy for ODet and ISeg across different model scales (*e.g.*, ViT-T/S/B/L (Li et al., 2021)), while only adding a small number of training parameters. Even with different training schedules (*i.e.*, 1×, and 3× with MS), our method can also improve the model performance, demonstrating the plug-and-play advantage of META. *ii*) Compared to the SOTA ViT-Adapter (Chen et al., 2022b) and LoSA (Mercea et al., 2024), META can achieve a new accuracy-efficiency trade-off. For example, in settings with the strong ViT-B (Li et al., 2021) as the backbone, our method achieves a performance gain of $0.5\%$AP$^m$ under 1× training schedule and $0.7\%$AP$^m$ under 3× training schedule with MS while reducing 4.9**M** model parameters, when compared to ViT-Adapter (Chen et al., 2022b). *iii*) Even with stronger pre-trained models, META still improves performance on the baseline models and surpasses existing methods on accuracy, parameters and memory. For example, on the ImageNet-22k pre-trained weights from (Steiner et al., 2021), our method achieves a performance gain of $0.8\%/0.7\%$AP$^m$ under 1×/3× training schedule while reducing 8.2**M** parameters compared to ViT-Adapter (Chen et al., 2022b), which validates its strong learning ability and flexible adaptability. *iv*) Compared to SOTA ODet and ISeg methods, META also has very competitive performance. For example, compared to the strong ViTDet-B (Li et al., 2022b) model, META-T has $2.2\%$AP$^b$ and $3.1\%$AP$^m$ gains under the 1× training schedule, and has META-T has $4.9\%$AP$^b$ and $2.7\%$AP$^m$ gains under the 3× training schedule. *v*) Compared to SOTA ViT adapter methods such as AdaptFormer (Chen et al., 2022a), FacT-TK (Jie & Deng, 2023), and LoSA (Mercea et al., 2024), our method exhibits superior efficiency in terms of reduced parameter count, decreased FLOPs, and lower MC. Particularly, META only utilizes $62\%$ of the MC of LoSA while achieving superior prediction accuracy. Therefore, the experimental results and conclusions above demonstrate that our method achieves better accuracy and higher efficiency in dense prediction tasks.

**Superiority performance under different baselines.** In addition to Mask R-CNN, we also choose Cascade Mask R-CNN (Cai & Vasconcelos, 2019), ATSS (Zhang et al., 2020), and GFL (Li et al., 2020) as the baselines as in Chen et al. (2022b); Mercea et al. (2024). We explore the effectiveness and superiority performance of META on these baseline models, where the 3× training with MS

| Methods | #P (M) | FLOPs (G) | MC (GB) | 1× schedule | | | | 3× +MS schedule | | | |
|---|---|---|---|---|---|---|---|---|---|---|---|
| | | | | $AP^b$ | $AP^m$ | $AP^m_{50}$ | $AP^m_{75}$ | $AP^b$ | $AP^m$ | $AP^m_{50}$ | $AP^m_{75}$ |
| ViT-T (Li et al., 2021) | 26.1 | 205 | NA | 35.5 | 33.5 | 54.9 | 35.1 | 44.9 | 37.0 | 59.6 | 39.0 |
| ViTDet-T (Li et al., 2022b) | 26.6 | 209 | NA | 35.7 | 33.5 | 54.7 | 35.2 | 40.4 | 37.1 | 60.1 | 39.3 |
| ViT-Adapter-T (Chen et al., 2022b) | 28.1 | 208 | 15.2 | 41.1 | 37.5 | 59.7 | 39.9 | 46.0 | 41.0 | 64.4 | 44.1 |
| AdaptFormer-T (Chen et al., 2022a) | 27.9 | 210 | 17.7 | 36.4 | 34.2 | 55.6 | 35.9 | 45.2 | 37.5 | 60.5 | 40.8 |
| FacT-TK-T (Jie & Deng, 2023) | 27.7 | 213 | 16.1 | 36.6 | 34.5 | 55.5 | 36.7 | 46.3 | 41.2 | 64.7 | 44.0 |
| LoSA-T (Mercea et al., 2024) | 27.6 | 208 | 13.0 | 41.4 | 38.0 | 60.1 | 40.5 | 46.1 | 39.5 | 64.2 | 44.3 |
| **META-T**(Ours) | 27.4 | 206 | 8.1 | 43.1 | 38.6 | 60.8 | 41.1 | 47.5 | 41.6 | 65.0 | 44.8 |
| ViT-S (Li et al., 2021) | 43.8 | 245 | NA | 40.2 | 37.1 | 59.9 | 39.3 | 44.0 | 39.9 | 63.4 | 42.2 |
| ViTDet-S (Li et al., 2022b) | 43.8 | 255 | NA | 40.6 | 37.1 | 60.0 | 38.8 | 44.5 | 40.1 | 63.6 | 42.5 |
| ViT-Adapter-S (Chen et al., 2022b) | 47.8 | 251 | 15.2 | 44.7 | 39.9 | 62.5 | 42.8 | 48.2 | 42.8 | 66.4 | 45.9 |
| AdaptFormer-S (Chen et al., 2022a) | 47.5 | 255 | 17.7 | 41.2 | 37.8 | 60.3 | 41.0 | 45.0 | 41.2 | 64.2 | 43.4 |
| FacT-TK-S (Jie & Deng, 2023) | 46.8 | 259 | 16.1 | 41.5 | 37.8 | 60.7 | 41.2 | 45.2 | 41.8 | 65.0 | 42.9 |
| LoSA-S (Mercea et al., 2024) | 46.1 | 249 | 13.0 | 44.2 | 39.8 | 62.6 | 42.5 | 48.3 | 42.5 | 66.5 | 46.0 |
| **META-S**(Ours) | 45.2 | 247 | 8.1 | 45.5 | 40.5 | 63.4 | 43.5 | 49.5 | 43.3 | 66.9 | 46.6 |
| ViT-B (Li et al., 2021) | 113.6 | 719 | NA | 42.9 | 39.4 | 62.6 | 42.0 | 45.8 | 41.3 | 65.1 | 44.4 |
| ViTDet-B (Li et al., 2022b) | 121.3 | 800 | NA | 43.2 | 39.2 | 62.7 | 41.4 | 46.3 | 41.6 | 65.3 | 44.5 |
| ViT-Adapter-B (Chen et al., 2022b) | 120.2 | 730 | 15.2 | 47.0 | 41.8 | 65.1 | 44.9 | 49.6 | 43.6 | 67.7 | 46.9 |
| AdaptFormer-B (Chen et al., 2022a) | 118.5 | 733 | 17.7 | 44.5 | 40.3 | 63.1 | 43.0 | 46.2 | 42.4 | 66.2 | 45.7 |
| FacT-TK-B (Jie & Deng, 2023) | 118.0 | 735 | 16.1 | 44.9 | 41.2 | 64.1 | 43.2 | 47.6 | 43.2 | 67.5 | 46.3 |
| LoSA-B (Mercea et al., 2024) | 117.2 | 722 | 13.0 | 45.1 | 41.8 | 64.6 | 44.0 | 48.6 | 43.8 | 67.9 | 47.0 |
| **META-B**(Ours) | 115.3 | 720 | 8.1 | 45.4 | 42.3 | 66.0 | 45.7 | 51.2 | 44.3 | 68.2 | 47.5 |
| ViT-L‡ (Li et al., 2021) | 337.3 | 1,907 | NA | 45.7 | 41.5 | 65.6 | 44.6 | 48.3 | 43.4 | 67.9 | 46.6 |
| ViTDet-L‡ (Li et al., 2022b) | 350.9 | 1,900 | NA | 46.2 | 41.4 | 65.8 | 44.1 | 49.1 | 44.0 | 68.5 | 47.6 |
| ViT-Adapter-L‡ (Chen et al., 2022b) | 347.9 | 1,927 | 15.2 | 48.7 | 43.3 | 67.0 | 46.9 | 52.1 | 46.0 | 70.5 | 49.7 |
| AdaptFormer-L‡ (Chen et al., 2022a) | 345.7 | 1,924 | 17.7 | 46.3 | 42.6 | 66.1 | 45.8 | 49.0 | 44.8 | 68.1 | 47.1 |
| FacT-TK-L‡ (Jie & Deng, 2023) | 342.1 | 1,925 | 16.1 | 46.5 | 42.9 | 66.5 | 46.0 | 48.8 | 44.5 | 68.2 | 47.0 |
| LoSA-L‡ (Mercea et al., 2024) | 341.3 | 1,922 | 13.0 | 50.2 | 43.5 | 66.8 | 47.1 | 53.2 | 46.2 | 71.0 | 48.9 |
| **META-L**‡(Ours) | 339.7 | 1,905 | 8.1 | 51.8 | 44.1 | 67.6 | 47.8 | 55.3 | 46.7 | 71.3 | 50.4 |

Table 1: Result comparisons with SOTA methods under Mask R-CNN (He et al., 2017) on the *val* set of MS-COCO. ‡ denotes the model is pre-trained on ImageNet-22k as in (Steiner et al., 2021). "MS" denotes the multi-scale training strategy. "NA" denotes not applicable.

strategy is used. The experimental results are given in Table 2. We can observe that META can consistently improve performance across different baselines, and exhibits more accurate and more efficient advantages compared to SOTA ViT adapter methods. For example, based on the Cascade Mask R-CNN (Cai & Vasconcelos, 2019), META-S can achieve up to $44.8\%$ $AP^m$ with only 83**M** parameters, 797**G** FLOPs, and 8.1**GB** MC, which is $1.3\%$ $AP^m$ higher, 3**M** fewer parameters, 4**G** fewer FLOPs, and 7.1**GB** fewer MC compared to the competitive ViT-Adapter-T method (Chen et al., 2022b). Besides, META-B can achieve a $1.7\%$ $AP^b$/$0.5\%$ $AP^m$ gain on the 3× training schedule, with 5**M** fewer parameters and 5**G** fewer FLOPs than the ViT-Adapter-B (Chen et al., 2022b) model. On ATSS and GFL, our method also achieves competitive $43.2\%$ and $44.0\%$ $AP^m$, respectively, which are $0.7\%$ and $0.9\%$ $AP^m$ higher than the competitive vision adapter ViT-Adapter-S (Chen et al., 2022b). In terms of efficiency, compare with the SOTA LoSA (Mercea et al., 2024) method, our method has fewer parameters (33**M** v.s. 35**M**), fewer FLOPs (265**G** v.s. 268**G** under ATSS, 279**G** v.s. 284**G** under GFL), and fewer MC (8.1**GB** v.s. 13.0**GB** under ATSS and GFL), indicating its higher computation and memory efficiency.

### 4.3 EXPERIMENTS ON SEMANTIC SEGMENTATION (SSEG)

**Baselines and Settings.** Following (Chen et al., 2022b; Jie & Deng, 2023; 2022), we select Semantic FPN (Kirillov et al., 2019) and UperNet (Xiao et al., 2018) as baseline models, where the Semantic FPN is trained for 80k iterations and the UperNet is trained for 160k iterations as in (Wang et al., 2021; Liu et al., 2021). Unless otherwise specified, the training and inference settings are set up to be consistent with the ViT-Adapter (Chen et al., 2022b).

| Methods | #P | FLOPs | MC | FPS | $AP^b$ | $AP^m$ | $AP^m_{50}$ | $AP^m_{75}$ |
|---|---|---|---|---|---|---|---|---|
| Cascade Mask R-CNN (Cai & Vasconcelos, 2019) 3× +MS schedule | | | | | | | | |
| Swin-T (Liu et al., 2021) | 86 | 745 | NA | 15.3 | 50.5 | 43.7 | 66.6 | 47.1 |
| ViT-S (Li et al., 2021) | 82 | 795 | NA | 16.5 | 47.9 | 42.8 | 62.1 | 44.8 |
| ViT-Adapter-S (Chen et al., 2022b) | 86 | 801 | 15.2 | 13.1 | 51.5 | 43.5 | 63.0 | 48.2 |
| LoSA-S (Mercea et al., 2024) | 84 | 799 | 13.0 | 12.0 | 53.5 | 43.8 | 64.1 | 48.1 |
| **META-S**(Ours) | 83 | 797 | 8.1 | 14.7 | 54.3 | 44.8 | 66.8 | 49.5 |
| Swin-B (Liu et al., 2021) | 145 | 982 | NA | 11.6 | 51.9 | 45.0 | 68.4 | 48.7 |
| ViT-B (Li et al., 2021) | 151 | 1,100 | NA | 13.0 | 50.1 | 42.7 | 63.9 | 44.1 |
| ViT-Adapter-B (Chen et al., 2022b) | 158 | 1,106 | 15.2 | 8.6 | 52.1 | 43.6 | 64.3 | 45.2 |
| LoSA-B (Mercea et al., 2024) | 155 | 1,104 | 13.0 | 9.0 | 52.6 | 43.5 | 63.8 | 44.7 |
| **META-B**(Ours) | 153 | 1,101 | 8.1 | 11.9 | 53.8 | 44.1 | 65.2 | 45.6 |
| ATSS (Zhang et al., 2020) 3× +MS schedule | | | | | | | | |
| Swin-T (Liu et al., 2021) | 36 | 215 | NA | 17.1 | 47.2 | 41.2 | 54.8 | 45.5 |
| ViT-S (Li et al., 2021) | 32 | 263 | NA | 18.0 | 45.2 | 40.5 | 52.0 | 41.8 |
| ViT-Adapter-S (Chen et al., 2022b) | 36 | 272 | 15.2 | 14.2 | 49.6 | 42.5 | 55.1 | 46.5 |
| LoSA-S (Mercea et al., 2024) | 35 | 268 | 13.0 | 16.0 | 50.3 | 41.9 | 54.4 | 45.0 |
| **META-S**(Ours) | 33 | 265 | 8.1 | 16.8 | 54.9 | 43.2 | 55.6 | 47.4 |
| GFL (Li et al., 2020) 3× +MS schedule | | | | | | | | |
| Swin-T (Liu et al., 2021) | 36 | 251 | NA | 17.5 | 47.6 | 40.8 | 54.2 | 46.0 |
| ViT-S (Li et al., 2021) | 32 | 275 | NA | 18.1 | 46.0 | 40.2 | 52.9 | 45.1 |
| ViT-Adapter-S (Chen et al., 2022b) | 36 | 288 | 15.2 | 15.3 | 50.0 | 43.1 | 54.5 | 47.1 |
| LoSA-S (Mercea et al., 2024) | 35 | 284 | 13.0 | 16.2 | 51.2 | 43.3 | 54.7 | 47.5 |
| **META-S**(Ours) | 33 | 279 | 8.1 | 17.6 | 55.6 | 44.0 | 55.2 | 48.5 |

Table 2: Result comparisons with SOTA methods under Cascade Mask R-CNN (Cai & Vasconcelos, 2019), ATSS (Zhang et al., 2020), and GFL (Li et al., 2020) on the *val* set of MS-COCO.

**Comparisons with state-of-the-art methods.** We show the SSeg experimental results of META under different settings and compare them with SOTA SSeg methods in Table 3. We can observe that *i*) Our method can consistently improve performance across different baselines, model scales, training strategies, and pre-training weights (including IN-1K, IN-22K, and MM), while having fewer model parameters and memory consumption compared to the advanced ViT-Adapter (Chen et al., 2022b) and LoSA (Mercea et al., 2024) methods. This demonstrates the strong applicability and learning ability of META, which can solve the problem of limited applicability in certain scenarios of downstream models in the traditional pre-training and fine-tuning strategy at the application level. *ii*) Compared to SOTA methods, META also can achieve a new SOTA accuracy-cost trade-off. The parameter from META is only about 0.2% of the overall model parameter count, indicating a minimal impact on the total parameter count. Furthermore, the memory consumption of our method accounts for only 55.14% of the SOTA LoSA (Mercea et al., 2024). *iii*) On larger model scales, META achieves higher efficiency with fewer parameters (*e.g.*, -1.4**M** on META-T and -4.2**M** on META-B). This indicates that our method is suitable for fine-tuning large ViT models.

## 4.4 ABLATION ANALYSIS

In our ablation analysis for ISeg and ODet, we adopt Mask R-CNN (He et al., 2017) under the 3× training schedule with MS as the baseline and report the experimental results on the *val* set of MS-COCO (Caesar et al., 2018), where the ImageNet-1k pre-trained ViT-B (Li et al., 2022b) is used as the backbone. For SSeg, we choose UperNet (Xiao et al., 2018) with 160k iterations as the baseline, where the ImageNet-1k pre-trained ViT-B (Li et al., 2022b) is used as the backbone. We report the single-scale testing results on the *val* set of ADE20K (Zhou et al., 2017). Due to page limitations, we only present the results of the effectiveness of each component in the main paper. Other aspects, such as result comparisons with other variants of ViT adapter methods, will be provided in the supplementary materials as part of the ablation analysis.

**Effectiveness of each component.** The experimental results of gradually adding META components on the pre-trained ViT-B (Li et al., 2022b) model are shown in Table 4. First, we can observe that directly using the Attn branch as the adapter on the pre-trained ViT-B (Li et al., 2022b) model not only fails to improve performance but also significantly decreases $AP^m$ and $AP^b$ (*i.e.*, from

| Methods | Pre-Training | Semantic FPN 80k | | | | UperNet 160k | | | |
|---|---|---|---|---|---|---|---|---|---|
| | | #P | MC | mIoU | mIoU$_{(w/MS)}$ | #P | MC | mIoU | mIoU$_{(w/MS)}$ |
| PVT-Tiny (Wang et al., 2021) | IN-1k | 17.0 | NA | 36.6 | 37.3 | 43.2 | NA | 38.5 | 39.0 |
| ViT-T (Li et al., 2021) | IN-1k | 10.2 | NA | 39.4 | 40.5 | 34.1 | NA | 41.7 | 42.6 |
| ViT-Adapter-T (Chen et al., 2022b) | IN-1k | 12.2 | 14.0 | 41.7 | 42.1 | 36.1 | 14.0 | 42.6 | 43.6 |
| LoSA-T (Mercea et al., 2024) | IN-1k | 11.1 | 10.7 | 40.8 | 41.9 | 36.3 | 10.7 | 42.1 | 43.7 |
| **META-T**$_{(Ours)}$ | IN-1k | 10.8 | 5.9 | 42.2 | 43.0 | 35.0 | 5.9 | 43.1 | 44.1 |
| Twins-SVT-S (Chu et al., 2021) | IN-1k | 28.3 | NA | 43.2 | 44.1 | 54.4 | NA | 46.2 | 47.1 |
| ViT-S (Li et al., 2021) | IN-1k | 27.8 | NA | 44.6 | 45.8 | 53.6 | NA | 44.6 | 45.7 |
| ViT-Adapter-S (Chen et al., 2022b) | IN-1k | 31.9 | 14.0 | 46.1 | 46.6 | 57.6 | 14.0 | 46.2 | 47.1 |
| LoSA-S (Mercea et al., 2024) | IN-1k | 29.8 | 10.7 | 45.7 | 46.7 | 54.9 | 10.7 | 45.8 | 46.6 |
| **META-S**$_{(Ours)}$ | IN-1k | 29.4 | 5.9 | 46.9 | 49.5 | 54.7 | 5.9 | 47.2 | 48.0 |
| Twins-SVT-L (Chu et al., 2021) | IN-1k | 103.7 | NA | 46.7 | 45.5 | 133.0 | NA | 48.8 | 50.2 |
| ViT-B (Li et al., 2021) | IN-1k | 98.0 | NA | 46.4 | 47.6 | 127.3 | NA | 46.1 | 47.1 |
| ViT-Adapter-B (Chen et al., 2022b) | IN-1k | 104.6 | 14.0 | 47.9 | 48.9 | 133.9 | 14.0 | 48.8 | 49.7 |
| LoSA-B (Mercea et al., 2024) | IN-1k | 103.5 | 10.7 | 48.3 | 49.2 | 131.2 | 10.7 | 47.3 | 48.5 |
| **META-B**$_{(Ours)}$ | IN-1k | 100.4 | 5.9 | 48.7 | 49.7 | 129.1 | 5.9 | 49.4 | 50.5 |
| Swin-L[‡] (Liu et al., 2021) | IN-22k | 234.0 | NA | – | – | 234.0 | NA | 52.1 | 53.5 |
| ViT-Adapter-B[‡] (Chen et al., 2022b) | IN-22k | 104.6 | 14.0 | 50.7 | 51.9 | 133.9 | 14.0 | 51.9 | 52.5 |
| ViT-Adapter-L[‡] (Chen et al., 2022b) | IN-22k | 332.0 | 14.0 | 52.9 | 53.7 | 363.8 | 14.0 | 53.4 | 54.4 |
| LoSA-L[‡] (Mercea et al., 2024) | IN-22k | 310.5 | 10.7 | 52.3 | 54.0 | 338.5 | 10.7 | 53.0 | 54.5 |
| **META-L**[‡]$_{(Ours)}$ | IN-22k | 307.7 | 5.9 | 53.7 | 54.5 | 336.3 | 5.9 | 54.0 | 55.2 |
| ViT-Adapter-L[*] (Chen et al., 2022b) | MM | 332.0 | 14.0 | 54.2 | 54.7 | 363.8 | 14.0 | 55.0 | 55.4 |
| LoSA-L[*] (Mercea et al., 2024) | MM | 310.1 | 10.7 | 53.3 | 54.5 | 341.0 | 10.7 | 55.1 | 55.5 |
| **META-L**[*]$_{(Ours)}$ | MM | 307.7 | 5.9 | 54.7 | 55.2 | 336.3 | 5.9 | 55.6 | 55.9 |

Table 3: Result comparisons with SOTA methods on the *val* set of ADE20K. "–" denotes there is no such a result in its paper. "IN" and "MM" denotes ImageNet and Multi-Modal, respectively.

| ISeg and ODet results on the *val* set of MS-COCO (Caesar et al., 2018) | | | | | | | | | | |
|---|---|---|---|---|---|---|---|---|---|---|
| ViT-B (Li et al., 2021) | Attn Branch. | FFN Branch. | Conv Branch. | Cascade | AP$^m$ | AP$^b$ | #P | FLOPs | MC | FPS |
| ✓ | | | | | 41.3 | 45.8 | 113.6 | 719 | NA | 11.5 |
| ✓ | ✓ | | | | 32.1 | 33.6 | 113.9 | 719 | 7.5 | 11.3 |
| ✓ | ✓ | ✓ | | | 43.4 | 46.5 | 114.4 | 719 | 7.5 | 11.3 |
| ✓ | | ✓ | | | 30.8 | 31.2 | 114.0 | 719 | 0.4 | 11.4 |
| ✓ | ✓ | ✓ | ✓ | | 44.1 | 48.1 | 115.3 | 719 | 7.5 | 10.2 |
| ✓ | ✓ | ✓ | | ✓ | 43.6 | 49.3 | 114.4 | 720 | 8.1 | 11.0 |
| ✓ | ✓ | ✓ | ✓ | ✓ | 44.3 | 51.2 | 115.3 | 720 | 8.1 | 11.1 |
| SSeg results on the *val* set of ADE20K (Zhou et al., 2017) | | | | | | | | | | |
| UperNet (Xiao et al., 2018) | Attn Branch. | FFN Branch. | Conv Branch. | Cascade | mIoU | | #P | FLOPs | MC | FPS |
| ✓ | | | | | 46.1 | | 127.3 | 1,841 | NA | 25.4 |
| ✓ | ✓ | | | | 33.7 | | 128.0 | 1,843 | 5.1 | 25.0 |
| ✓ | ✓ | ✓ | | | 46.8 | | 128.5 | 1,846 | 5.1 | 24.9 |
| ✓ | | ✓ | | | 31.1 | | 128.1 | 1,841 | 0.3 | 25.0 |
| ✓ | ✓ | ✓ | ✓ | | 47.9 | | 129.0 | 1,847 | 5.1 | 23.5 |
| ✓ | ✓ | ✓ | | ✓ | 48.0 | | 128.6 | 1,846 | 5.9 | 25.0 |
| ✓ | ✓ | ✓ | ✓ | ✓ | 49.4 | | 129.1 | 1,847 | 5.9 | 24.9 |

Table 4: Effectiveness of each component of META. "NA" denotes not applicable.

41.3% to 32.1% and from 45.8% to 33.6%). This indicates that using attention alone as the feature interaction method in the adapter is not enough. Then, on this basis, when we use both the Attn branch and FFN branch as adapter components (where the layer normalization is used as a shared manner), the model's accuracy is significantly improved, surpassing the baseline model by 2.1% AP$^m$/0.7% AP$^b$ and achieving 43.4% AP$^m$/46.5% AP$^m$. At the same time, the model only has a small increase in parameters and no increase in FLOPs. The same conclusion can be drawn from the SSeg task. *Furthermore, the independent utilization of the FFN branch as the adapter component does not yield an improvement in accuracy. Instead, it significantly detracts from the model's accuracy across all three tasks evaluated.* After that, continuing to add the Conv branch or using the

cascaded mechanism can bring sustained performance improvement, *e.g.*, 0.7% and 0.2% $AP^m$, and 1.6% and 2.8% $AP^b$, respectively. Finally, when using all components in the adapter, the model for ISeg and ODet achieves the best performance by 44.3% $AP^m$ and 51.2% $AP^b$ with 115.3**M** parameters, 720**G** FLOPs, 8.1**GB** MC, and 11.1 FPS. The model for SSeg achieves the best accuracy by 49.4 mIoU with 129.1**M** parameters, 1847**G** FLOPs, 5.9**GB** MC, and 24.9 FPS. In addition, results in the ablation analysis reveal that the increase in the model's FLOPs and MC is mainly attributed to the utilization of the "cascade" mechanism. Despite this, this mechanism can lead to significant improvements in both accuracy and speed.

## 5 CONCLUSION

We proposed a simple and fast META that aims at improving the ViT model's memory efficiency and reduce memory time consumption by minimizing layer normalization and frequent reshaping operations. The main contribution of this work was the proposal of the MEA block that shares the layer normalization between the self-attention and feed-forward network layers, which exist in a parallel manner. Within the proposed block, the cross-shaped self-attention was employed to reduce the requirements for frequent reshaping operations. Moreover, a lightweight local convolutional branch is introduced into META to enrich local inductive biases. Experimental results on object detection, instance segmentation, and semantic segmentation tasks demonstrated that META can achieve a new state-of-the-art accuracy-cost trade-off and at a faster inference speed. In addition to the experimental validation, we also theoretical proved that META exhibits superior generalization capability and stronger adaptability compared to current ViT adapters. In the future, we will explore the effectiveness of META on more ViT architectures, such as designing memory-efficient ViT models for a wider range of computer vision tasks. Besides, reducing the model's memory footprint is identified as a promising research direction to be explored for LLMs.

## ACKNOWLEDGMENT

The authors would like to thank all anonymous reviewers for their positive comments and constructive suggestions. This work was partially supported by the Hong Kong SAR RGC General Research Fund under Grant 16208823 and ACCESS - AI Chip Center for Emerging Smart Systems, sponsored by InnoHK funding, Hong Kong SAR. Rui Yan was supported by the Postdoctoral Fellowship Program of CPSF (Grant No. GZB20230302), Nanjing University Integrated Research Platform of the Ministry of Education - Top Talents Program (Grant No. 14380125).

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
