# Supplementary Materials for Paper:
# Memory Efficient Transformer Adapter for Dense Predictions

**Dong Zhang**[1,2]**, Rui Yan**[3]**, Pingcheng Dong**[1]**, Kwang-Ting Cheng**[1]
[1]The Hong Kong University of Science and Technology
[2]AI Chip Center for Emerging Smart Systems (ACCESS), [3]Nanjing University
`{dongz,timcheng}@ust.hk;ruiyan@nju.edu.cn;pingcheng.dong@connect.ust.hk`

In this supplementary material, we will provide a theoretical analysis to the proposed memory efficient Transformer adapter (META) in Section S1, provide a detailed description of the experimental datasets in Section S2, provide a detailed description of the experimental settings in Section S3, provide more result comparisons under different pre-trained weights in Section S4, provide more ablation study results in Section S5, show class activation map comparisons of instance segmentation before and after adding the Conv branch in Section S6, qualitative visualizations of instance segmentation and semantic segmentation results in Section S7, as well as the pseudo-code for when the stripe size is set to 2 in Section S8.

## S1    Theoretical Analysis of META

*This supplementary is for Section 3 of the main paper.* In this section, we will prove that META exhibits superior generalization capability and stronger adaptability compared to existing ViT adapters. To achieve this goal, we will prove that the proposed memory efficient adapter (MEA) block possesses larger information entropy (IE) than the existing attention-based ViT adapters (Hu et al., 2022; Jie & Deng, 2023; Chen et al., 2022; Ma et al., 2024; Luo et al., 2023; Shao et al., 2024), which provides evidence that the MEA block has more comprehensive feature representations. Then, based on the maximum mean discrepancy (MMD) theory (Cheng & Xie, 2021; Arbel et al., 2019; Wang et al., 2021a), larger IE in the ViT adapter framework leads to superior generalization capability and stronger adaptability. The detailed theoretical analysis process is as follows:

**Lemma S1.1.** *In any case of mutual information, the MEA block will gain larger information entropy after fusing $\mathbf{X}_{vit}$ and $\mathbf{X}_{con}$.*

*Proof.* As introduced in Section 3.2 of the main paper, the proposed MEA block can be viewed as an operation that integrates the ViT features (*i.e.*, the Attn branch and the FFN branch) and the convolution features (*i.e.*, the Conv branch). Therefore, we begin by formalizing the obtained features into the following two basic elements: the ViT features and the convolution features. To formalize the learning setting, we express the ViT features as $\mathbf{X}_{vit}$ and the convolution features as $\mathbf{X}_{con}$. It is evident that if $\mathbf{X}_{vit}$ and $\mathbf{X}_{con}$ are extracted from the same image, then $\mathbf{X}_{vit}$ and $\mathbf{X}_{con}$ are not independently distributed, and there exists some mutual information between them (Zhang et al., 2022; Wu et al., 2021; Zhang et al., 2023; Peng et al., 2021). Therefore, the IE of the fused feature of $\mathbf{X}_{vit}$ and $\mathbf{X}_{con}$ within the MEA block can be expressed as:

$$H(\mathbf{X}_{vit}, \mathbf{X}_{con}) = H(\mathbf{X}_{vit}) + H(\mathbf{X}_{con}) - I(\mathbf{X}_{vit}; \mathbf{X}_{con}), \tag{1}$$

where $H(\cdot)$ is utilized to calculate the IE of the given variate, which can be formulated as:

$$H(\mathbf{X}_{vit}) = -\sum P(\mathbf{x}_{vit}) log(P(\mathbf{x}_{vit})),$$
$$H(\mathbf{X}_{con}) = -\sum P(\mathbf{x}_{con}) log(P(\mathbf{x}_{con})), \tag{2}$$

where $P(\mathbf{x}_{vit})$ represents the probability of $\mathbf{X}_{vit}$ taking on the value of $\mathbf{x}_{vit}$. The similar definition of $P(\mathbf{x}_{con})$. $I(\cdot; \cdot)$ in Eq. equation 1 is used to compute the mutual information between $\mathbf{X}_{vit}$ and $\mathbf{X}_{con}$, which can be expressed as:

$$I(\mathbf{X}_{vit}; \mathbf{X}_{con}) = \sum \sum P(\mathbf{X}_{vit}, \mathbf{X}_{con}) log(P(\mathbf{X}_{vit}, \mathbf{X}_{con})(P(\mathbf{X}_{vit}), P(\mathbf{X}_{con}))), \tag{3}$$

where $P(\mathbf{X}_{vit}, \mathbf{X}_{con})$ is their joint probability distribution. Since $I(\mathbf{X}_{vit}; \mathbf{X}_{con})$ is always non-negative, $H(\mathbf{X}_{vit}, \mathbf{X}_{con})$ may still be greater than $H(\mathbf{X}_{vit})$ or $H(\mathbf{X}_{con})$ (Paninski, 2003; Gabrié et al., 2018). This suggests that the IE of the features extracted by MEA is always greater than the feature representation extracted by either of them separately.

Specifically, if $I(\mathbf{X}_{vit}; \mathbf{X}_{con})$ is small, the IE gain after fusion may still be significant, which is beneficial for improving the generalization capability and adaptability of the block. However, when $I(\mathbf{X}_{vit}; \mathbf{X}_{con})$ is large, the IE gain after fusion may be reduced. This means that $I(\mathbf{X}_{vit}; \mathbf{X}_{con})$ may affect the IE improvement of the fused model. Next, we will discuss the impact of $\mathbf{X}_{vit}$ and $\mathbf{X}_{con}$ on improving the IE of the adapter based on the size of $I(\mathbf{X}_{vit}; \mathbf{X}_{con})$, which can be divided into the following three cases:

- Small $I(\mathbf{X}_{vit}; \mathbf{X}_{con})$. This is an ideal state. When the dependency between $\mathbf{X}_{vit}$ and $\mathbf{X}_{con}$ is small, it indicates that $I(\mathbf{X}_{vit}; \mathbf{X}_{con})$ is small, that is, $\mathbf{X}_{vit}$ and $\mathbf{X}_{con}$ respectively represent different information of the image. In this case, fusing $\mathbf{X}_{vit}$ and $\mathbf{X}_{con}$ can bring a significant increase in IE, which is beneficial to improving the adapter's generalization capability and adaptability.

- Medium $I(\mathbf{X}_{vit}; \mathbf{X}_{con})$. When $I(\mathbf{X}_{vit}; \mathbf{X}_{con})$ is between small and large, it indicates that there is a certain degree of correlation between them. In this case, fusing $\mathbf{X}_{vit}$ and $\mathbf{X}_{con}$ may still bring some IE gain. The specific improvement effect depends on the degree of correlation between $\mathbf{X}_{vit}$ and $\mathbf{X}_{con}$ and their complementarity in image representations. Fortunately (Zhang et al., 2022; 2023; Marouf et al., 2024; Liu et al., 2023), a large amount of work has validated that ViT and convolutional layers can extract distinctive information from images. Therefore, in this case, fusing $\mathbf{X}_{vit}$ and $\mathbf{X}_{con}$ can still bring IE gains.

- Large $I(\mathbf{X}_{vit}; \mathbf{X}_{con})$. When $I(\mathbf{X}_{vit}; \mathbf{X}_{con})$ between $\mathbf{X}_{vit}$ and $\mathbf{X}_{con}$ is large, it indicates that there is a high correlation between them, *i.e.*, global ViT and local convolution features may represent similar or overlapping information of the image. In this case, the IE gain brought by fusing $\mathbf{X}_{vit}$ and $\mathbf{X}_{con}$ may decrease because there is a lot of information overlap between them. However, in our case, the probability of such a scenario occurring is almost non-existent, fusing $\mathbf{X}_{vit}$ and $\mathbf{X}_{con}$ may still improve the performance of the model to some extent, because they may capture the detailed information of the image to varying degrees.

Based on the aforementioned theoretical analysis, we can conclude that the proposed MEA block has a larger IE than existing ViT adapters (which are primarily based on the attention mechanism) under any scenario. This provides evidence that the MEA block has more comprehensive feature representations. □

As the MEA block includes a parallel convolutional branch, it can better capture local inductive biases compared to the traditional ViT adapter, which mainly uses self-attention (Hu et al., 2022; Jie & Deng, 2023; Chen et al., 2022; Ma et al., 2024; Luo et al., 2023; Shao et al., 2024; Mercea et al., 2024). Therefore, the MEA block's feature space should be more capable of distinguishing different samples, resulting in a larger MMD value. Our MEA block's feature space is obtained by combining the attention branch, the feed-forward network branch, and the local convolutional branch, enabling it to capture both local and global inductive biases of the given image. In contrast, the traditional ViT adapter's feature space is mainly obtained through self-attention and may not be able to capture local features well. Therefore, according to the MMD theory (Cheng & Xie, 2021; Arbel et al., 2019; Wang et al., 2021a), we can conclude that if the MEA block's feature space is more discriminative than the traditional ViT adapter's feature space, then the MEA block's feature space is more suitable for adapter feature space and can better improve the model's generalization capability and adaptability.

## S2 INTRODUCTION OF THE EXPERIMENTAL DATASETS

*This supplementary is for Section 4.1 of the main paper.* In our paper, two representative datasets are used to evaluate the effectiveness and efficiency of our method, including MS-COCO (Caesar et al., 2018) for ODet and ISeg, and ADE20K (Zhou et al., 2017) for SSeg. Below are the details of the used datasets:

| Methods | Pre-Trained | Params.↓ | $AP^m$ ↑ |
|---|---|---|---|
| Swin-B (Liu et al., 2021) | ImageNet-1k (Deng et al., 2009) | 107.1 | 43.3 |
| ViT-Adapter-B (Chen et al., 2022) | ImageNet-1k (Deng et al., 2009) | 120.2 | 43.6 |
| **META-B**(Ours) | ImageNet-1k (Deng et al., 2009) | 115.3 | $44.3_{+0.7}$ |
| Swin-B (Liu et al., 2021) | ImageNet-22k (Steiner et al., 2021) | 107.1 | 44.3 |
| ViT-Adapter-B (Chen et al., 2022) | ImageNet-22k (Steiner et al., 2021) | 120.2 | 44.6 |
| **META-B**(Ours) | ImageNet-22k (Steiner et al., 2021) | 115.3 | $45.2_{+0.6}$ |
| Swin-B (Liu et al., 2021) | Multi-Modal (Zhu et al., 2022) | 107.1 | – |
| ViT-Adapter-B (Chen et al., 2022) | Multi-Modal (Zhu et al., 2022) | 120.2 | 45.3 |
| **META-B**(Ours) | Multi-Modal (Zhu et al., 2022) | 115.3 | $45.9_{+0.6}$ |

Table S1: Result comparisons on Params. (**M**) and AP (%) under different pre-trained weights with Mask R-CNN (3× +MS schedule) (He et al., 2017) as the baseline model on the *val* set of MS-COCO (Caesar et al., 2018). "–" denotes there is no such a result in its paper.

- MS-COCO (Caesar et al., 2018) is a representative yet challenging dataset for common scene IS and object detection, which consists of 118k, 5k and 20k images for the *training* set, the *val* set and the *test* set, respectively. In our experiments, the model is trained on the *training* set and evaluated on the *val* set.
- ADE20K (Zhou et al., 2017) is a scene parsing dataset with 20k images and 150 object categories. Each image has pixel-level annotations for SS of objects and regions within the scene. The dataset is divided into 20k, 2k, and 3k images for *training*, *val* and *test*, respectively. Our model is trained on the *training* set and evaluated on the *val* set.

For data augmentation, random horizontal flip, brightness jittering and random scaling within the range of $[0.5, 2]$ are used in training as in (Chen et al., 2022; Luo et al., 2023; Zhang et al., 2023; Mercea et al., 2024). By default, the inference results are obtained at a single scale, unless explicitly specified otherwise.

## S3   INTRODUCTION OF THE EXPERIMENTAL SETTINGS

*This supplementary is for Section 4.2 of the main paper.* Experiments on object detection and instance segmentation are conducted using the open-source MMDetection framework (Chen et al., 2019). The training batch size is set to 16, and AdamW (Loshchilov & Hutter, 2017) is used as the optimizer with the initial learning rate of $1 \times 10^{-4}$ and the weight decay of 0.05. The layer-wise learning rate decay is used and set to 0.9, and the drop path rate is set to 0.4. Following (Xiong et al., 2024; Wang et al., 2021b; Chen et al., 2022; Liu et al., 2022), to ensure a fair result comparison, we choose two training schedules, 1× (*i.e.*, 12 training epochs) and 3× (*i.e.*, 36 training epochs). For the 1× training schedule, images are resized to the shorter side of 800 pixels, with the longer side not exceeding $1,333$ pixels. In inference, the shorter side of images is consistently set to 800 pixels by default. For the 3× training schedule, the multi-scale training strategy is also used as in (Chen et al., 2022), and the shorter side is resized to 480 to 800 pixels, while the longer side remains capped at $1,333$ pixels.

*This supplementary is for Section 4.3 of the main paper.* Experiments on semantic segmentation are conducted using the MMSegmentation framework (Contributors, 2020). The input images are cropped to a fix size of $512 \times 512$ pixels as in (Xiong et al., 2024; Chen et al., 2022). The training batch size is set to 16, and AdamW (Loshchilov & Hutter, 2017) is used as the optimizer with the initial learning rate of $1 \times 10^{-5}$ and the weight decay of 0.05. Following (Li et al., 2022; Liu et al., 2021), the layer-wise learning rate decay is set to 0.9 and the drop path rate is set to 0.4. We report the experimental results on both single scale training and multi-scale training strategies.

## S4   RESULT COMPARISONS UNDER DIFFERENT WEIGHTS

*This supplementary is for Section 4.2 of the main paper.* In this section, we present the experimental results of META on object detection and instance segmentation with different pre-trained weights

| Settings | ViT-B | Attn | FFN | Conv | Cascade | AP$^m$ ↑ | FPS↑ | Params.↓ | FLOPs↓ | MC↓ |
|---|---|---|---|---|---|---|---|---|---|---|
| Baseline model | ✓ | ✗ | ✗ | ✗ | ✗ | 41.3 | 11.5 | 113.6**M** | 719**G** | NA |
| Shared normalization | ✓ | ✓ | ✓ | ✗ | ✗ | 43.4 | 11.3 | 114.4**M** | 719**G** | 7.5**GB** |
| Non-shared normalization | ✓ | ✓ | ✓ | ✗ | ✗ | 43.2 | 10.5 | 114.4**M** | 737**G** | 8.8**GB** |

Table S2: Ablation study results on shared layer normalization.

| Methods | AP↑ | FPS↑ | Params. (**M**)↓ | FLOPs (**G**)↓ | Momory (**GB**)↓ |
|---|---|---|---|---|---|
| WindowAtt (Liu et al., 2021) | 41.2 | 11.6 | 145.0 | 982 | 18.5 |
| PaleAttention (Wu et al., 2022) | 42.8 | 14.4 | 155.2 | 1,029 | 16.7 |
| Attention (Vaswani et al., 2017) | 43.1 | 5.2 | 188.4 | 1,250 | 18.3 |
| CSWindow (Dong et al., 2022) | 43.1 | 13.7 | 144.6 | 990 | 12.9 |
| SimplingAtte (He & Hofmann, 2024) | 43.3 | 12.2 | 126.3 | 994 | 17.1 |
| DeformableAtt (Xia et al., 2022) | 43.7 | 13.5 | 166.0 | 988 | 15.2 |
| MiniAdapters (Marouf et al., 2024) | 41.9 | 15.0 | 131.8 | 995 | 12.2 |
| VL-Adapter (Sung et al., 2022) | 42.7 | 14.5 | 167.2 | 993 | 14.0 |
| **META-B**(Ours) | 44.3 | 17.4 | 115.3 | 720 | 8.1 |

Table S3: Result comparisons with different adapters.

and compare them with other state-of-the-art methods including SwinViT (Liu et al., 2021) and ViT-Adapter (Chen et al., 2022) as in (Chen et al., 2022). Mask R-CNN (He et al., 2017) is used as the baseline, and ViT-B (Li et al., 2022) is used as the backbone. The 3× training schedule with MS training strategy is used. The obtained experimental results are given in Table S1. From this table, we can observe that our method is applicable to different pre-trained weights (*i.e.*, ImageNet-1k (Deng et al., 2009), ImageNet-22k (Steiner et al., 2021), and Multi-Modal (Zhu et al., 2022)), and achieves more accurate AP with fewer model parameters compared to ViT-Adapter (Chen et al., 2022), across different pre-trained weights.

## S5 MORE ABLATION STUDY RESULTS

*This supplementary is for Section 4.4 of the main paper.* In our main paper, we present the experimental results of deploying adapters with Attn branch and FFN branch as components on ViT-B (Li et al., 2022). It is noteworthy that the layer normalization operation has been shared between the Attn branch and the FFN branch to reduce the memory access costs associated with the normalization operations. In this section, we demonstrate a result comparison between the experimental results of using shared layer normalization operation and those of not using it in the traditional setting (*i.e.*, the non-shared normalization). The obtained experimental results are shown in Table S2. It can be observed that sharing layer normalization does not significantly improve the performance in terms of AP. However, compared to FPS, FLOPs, MC, our approach can achieve satisfactory performance gains.

*This supplementary is for Section 4.4 of the main paper.* META is proposed as a simple and fast ViT adapter by minimizing inefficient memory access operations. In this section, we compare META with other efficient attention methods and advanced adapter methods (Marouf et al., 2024; Xia et al., 2022; Sung et al., 2022). All methods are used with their default settings and the same settings as the injector and extractor in ViT-adapter (Chen et al., 2022). Following the same setup as in (Chen et al., 2022), the attention mechanism is utilized as the ViT-adapter layer. Therefore, during the experimental comparisons, we replace the attention mechanism in the ViT-adapter with alternative attention mechanisms to ensure a fair comparison. The obtained experimental results are given in Table S3. We can observe that compared to these methods, META achieves new state-of-the-art performance in both accuracy and efficiency. We ultimately achieve an AP of 44.3% with 115.3**M** parameters, 720**G** FLOPs, 17.4 FPS, and 8.1 **GB** MC.

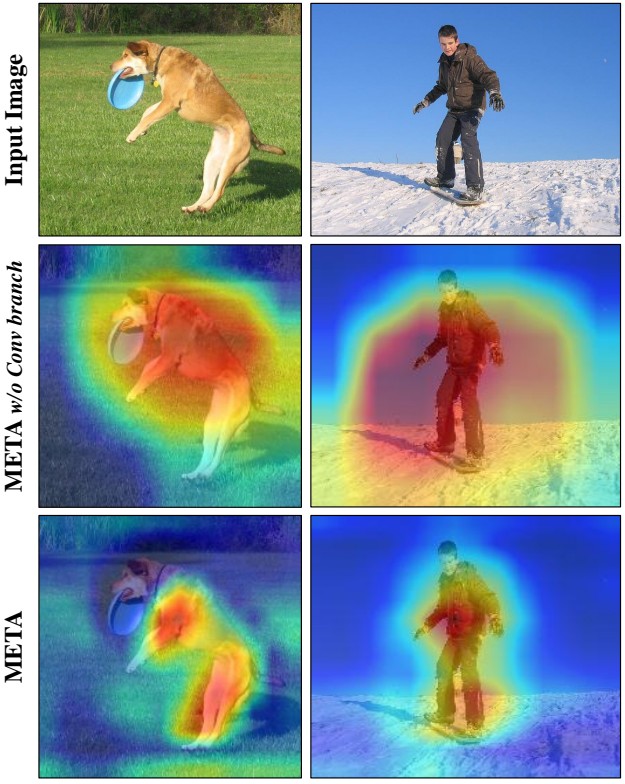

Figure S1: Class activation map comparisons of instance segmentation before and after adding the Conv branch. The sample images are from the *training* set of MS-COCO (Caesar et al., 2018).

## S6 VISUALIZATIONS UNDER THE CONV BRANCH

*This supplementary is for Section 3.2 of the main paper.* In this section, to observe if the adapter has learned local inductive biases through the Conv branch, we visualize the model's class activation maps. The obtained visualizations are given in Figure S1. From this figure, it can be observed that after adding the Conv branch, the model focuses more on the specific object area (*e.g.*,," the dog" and "the person") rather than the surrounding area that may extend beyond the object itself, as was the case before adding the Conv branch. This indicates that our method effectively learns local inductive biases after incorporating the Conv branch.

## S7 QUALITATIVE VISUALIZATION RESULTS

*This supplementary is for Section 4.2 and 4.3 of the main paper.* In this section, we show qualitative results on both instance segmentation and semantic segmentation. To demonstrate the superiority of our method, we present visualization results of ablation studies on instance segmentation, as well as comparisons with state-of-the-art methods on both instance segmentation and semantic segmentation. The obtained visualization results are shown in Figure S2. From the results, it can be observed that compared to other methods, our method can achieve more accurate object masks that better fit the actual boundaries of the objects themselves.

as well as the pseudo-code for when the stripe size is set to 2 in Section S8.

## S8 PSEUDO-CODE FO STRIPE SIZE $= 2$

In this code snippet, stripe size is set to 2, and relevant features are directly obtained using the gather function instead of reshaping them with img2windows. This operation can reduce unnecessary reshaping operations and improves the efficiency of the code.

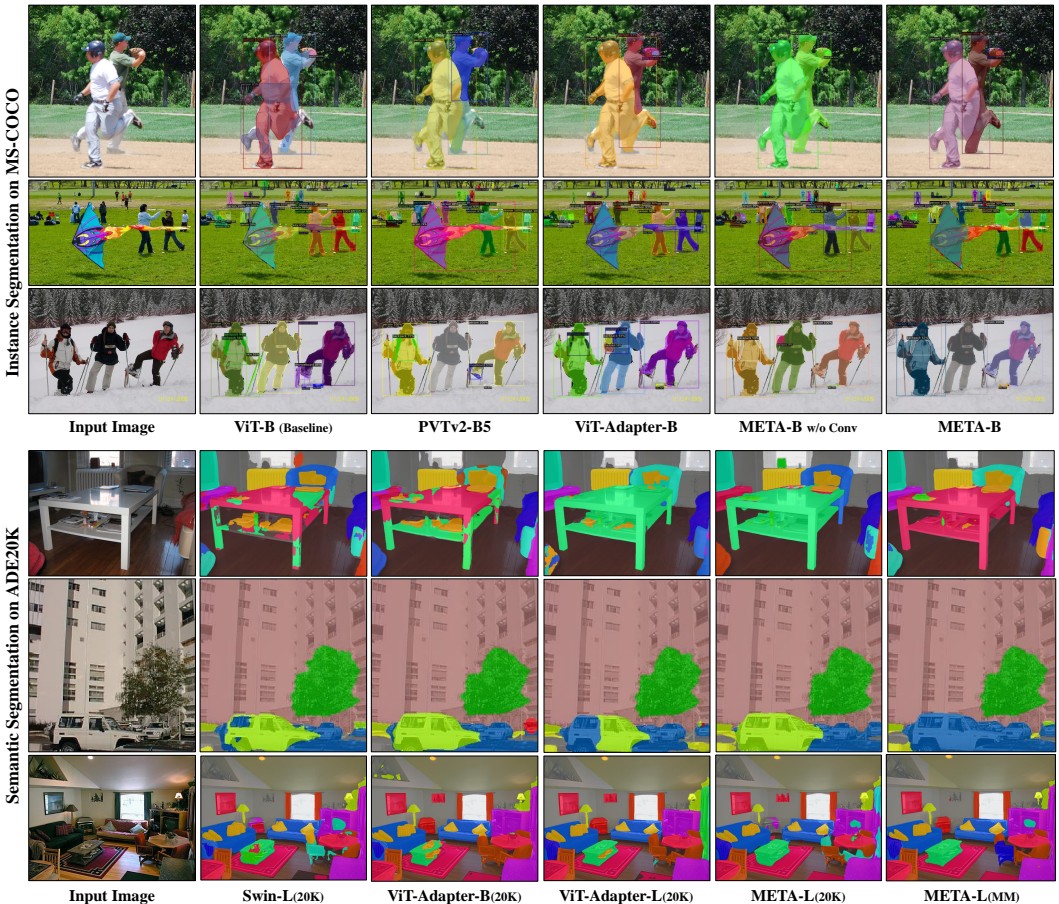

Figure S2: Qualitative results. The sample images are from the *val* set of MS-COCO (Caesar et al., 2018) for instance segmentation, and are from the *val* set of ADE20K (Zhou et al., 2017) for semantic segmentation. "w/o Conv" denotes that the Conv branch is not used in the experiments. "20K" and "MM" refers to the backbone network being pre-trained on ImageNet-22k (Steiner et al., 2021) and Multi-Modal (Zhu et al., 2022), respectively.

```
function cross_shaped_window_attention(x, num_heads, window_size):
    # x: given feature
    # num_heads: head number
    # window_size: window size

    # Get dimensions
    (batch_size, seq_length, d_model) = shape(x)

    # Split into multiple heads
    Q, K, V = split_heads(x, num_heads)

    # Initialize attention output
    attention_output = zeros(batch_size, seq_length, d_model)

    # Initialize previous head's output for cascaded attention
    previous_Q = zeros(batch_size, seq_length, d_model)
    previous_K = zeros(batch_size, seq_length, d_model)
    previous_V = zeros(batch_size, seq_length, d_model)

    # Calculate attention for each head
    for head in range(num_heads):
```

```
        for position in range(seq_length):
            # Get cross-shaped window indices
            window_indices = get_cross_shaped_window_indices(position,
                                              window_size)

            # Gather Q, K, V for the current window
            Q_window = gather(Q[head], window_indices)
            K_window = gather(K[head], window_indices)
            V_window = gather(V[head], window_indices)

            # Incorporate previous head's output for cascaded attention
            if head > 0:
                Q_window += previous_Q
                K_window += previous_K
                V_window += previous_V

            # Calculate attention scores
            attention_scores = softmax(Q_window * K_window^T / sqrt(d_k))

            # Compute the attention output for the current position
            attention_output[position] = attention_scores * V_window

        # Update previous head's output for the next head
        previous_Q = Q_window
        previous_K = K_window
        previous_V = V_window

    # Final linear transformation
    attention_output = linear_transform(attention_output)
    return attention_output

function feed_forward_network(x):
    # Feed Forward Network
    x = ReLU(linear(x))
    x = linear(x)
    return x
```

```
def get_cross_shaped_window_indices(position, window_size, seq_length):
    # Initialize the list of indices
    indices = []

    # Add the current position
    indices.append(position)

    # Add vertical neighbors (up and down)
    for offset in range(-window_size, window_size + 1):
        if position + offset >= 0 and position + offset < seq_length:
            indices.append(position + offset)

    # Remove duplicates and sort the indices
    indices = list(set(indices))
    indices.sort()

    return indices
```