# OpenReview forum: "Memory Efficient Transformer Adapter for Dense Predictions"
_ICLR.cc/2025/Conference — ICLR 2025 Poster_

### Official Review · Reviewer_oVw4 · 2024-10-27

**Soundness:** 2
**Presentation:** 2
**Contribution:** 3
**Rating:** 6
**Confidence:** 4

**Summary:**

This paper proposes a Memory-Efficient Transformer Adapter, termed META, which reduces memory access costs by sharing layer normalization across multiple modules and substituting standard self-attention with cross-shaped self-attention. Meanwhile, META divides the feature map into smaller parts along the channel dimension and processes these smaller features sequentially. Thereby further reducing memory requirements. Experiment results in object detection and instance segmentation indicate that META achieves better accuracies.

**Strengths:**

1. META introduces a cross-shaped self-attention mechanism and a cascaded process, both of which are grounded in the principles of dividing the entire feature into multiple smaller features to reduce memory costs.
2. META incorporates local inductive biases by introducing convolutions into the FFN and an additional lightweight convolutional branch. This enables META to achieve better performance in extensive experimental evaluations.

**Weaknesses:**

1. Insufficient Motivation (1): META claims that the inference speed of previous adapters is hindered by inefficient memory access operations such as normalization and frequent reshaping, but it lacks experimental analysis to support this claim. It is recommended to provide a detailed breakdown of inference time to show the proportion of inefficient memory access operations in META and previous methods.
2. Insufficient Motivation (2): META aims to decrease memory access costs by reducing frequent reshaping operations. However, I do not observe any reduction. First, the input for attention and layer normalization is $x\in R^{B\times L\times C}$, where B, L, and C denote the batch size, number of tokens, and channels, respectively. In contrast, the convolution accepts input in the format $x\in R^{B\times C\times H\times W}$. The MEA block mixes many convolutions, layer normalization, and attention. This may result in multiple tensor reshaping operations. Second, the cross-shaped self-attention mechanism divides the features into non-overlapping horizontal/vertical stripes, further compounding the need for tensor reshaping operations. I conjecture that the observed lower memory access costs during experiments are due to the segmentation of the entire feature into multiple smaller features, instead of reducing tensor reshaping operations. I'd like to see a thorough analysis of memory costs associated with each operation in META and previous approaches. This will help clarify where the memory saving comes from.
3. The results of the ablation study presented in Table 4 indicate that convolutional layers are primarily responsible for the observed improvements (FFN also includes MLP composed of two 3x3 convolutional layers). This raises the question: to what extent does the Attention Branch contribute to these improvements? Consider conducting an additional ablation study that includes the ViT-B along with the FFN Branch, maintaining the same configuration as described in Line 435, but excluding the Attn Branch.
4. The proposed META is relatively sophisticated and comprises numerous layers (e.g., the cascaded injector includes 16 layers), making it less practical for low-performance hardware. On which hardware do you measure FPS? It is recommended to compare META with other methods on less powerful GPUs such as the V100, rather than A100 or H100.
5. In Table S3, how do you compare other efficient attention methods? Do you only replace the attention mechanism in the ViT-adapter with other attention mechanisms? Please provide further details regarding the experimental setup.
6. Other minor comments.
Line 150：The spatial prior requires clarification, is the spatial prior module utilized here identical to that in the ViT-adapter [1]?
Line 96: TDE Transformer, DeiT is more frequently used.
Line 182: The term "which" appears to ambiguously refer to the prior module rather than the MEA block; it would be beneficial to provide clarification.
Line 188: In Equation 1, "Concat" is a widely recognized abbreviation for concatenation.
Line 199: "Attn" is a more commonly accepted abbreviation for attention compared to "Atte".
Line 223: Should "respectively" be replaced with "sequentially"?
Line 166 Table S2 in Supplementary Materials: Do you mean separate normalization for different modules? The use of "common" may introduce ambiguity.

[1] Zhe Chen, Yuchen Duan, Wenhai Wang, Junjun He, Tong Lu, Jifeng Dai, and Yu Qiao. Vision transformer adapter for dense predictions. In ICLR 2023.

**Questions:**

Please see weakness section. After the rebuttal, I have addressed many of my concerns and now support the acceptance of the paper.

---

> ### Author Response · Authors · 2024-11-18
> **Response to Reviewer oVw4 (part 1)**
>
> We appreciate your acknowledgment of our work in reducing memory costs while achieving enhanced performance through extensive experimental evaluations. The following is our response with some necessary explanations. If you have any further questions, please feel free to let us know.
>
> Q1: A detailed breakdown of inference time.
>
> A1: Thanks for your suggestion. The inference time analysis for each component of META is provided in Table 4 of our main paper, and the comparative evaluation of the inference time with previous methods is included in Table S3 of the supplementary materials. We use the frames per second (FPS) during the inference process as the metric to measure the model's inference time.
>
> From Table 4, we can observe that: a) The addition of the Attention branch and FFN branch does not significantly reduce FPS compared to the baseline model, indicating that these two branches do not substantially impede the model's inference time. b) The incorporation of the Conv branch leads to a slight decrease in FPS, suggesting that convolutional layers impact the model's speed due to the increased model complexity. c) The implementation of the cascaded strategy can further enhance the model's inference speed, as this manner allows for the parallel processing of different features, particularly with current GPUs support for parallel computation. Furthermore, the cascaded strategy can facilitate the rapid feature fusion, thus avoiding complex feature interactions and enhancing the inference efficiency.
>
> In Table S3 of the supplementary materials, we compare META with previous efficient attention methods, including Window Attention[R1], Pale Attention[R2], Dense Attention[R3], CSWindow[R4], SimpleAttention[R5], and Deformable Attention[R6]. We evaluate the model's accuracy, complexity, memory, and inference efficiency. The experimental results indicate that our method achieves state-of-the-art performance in both accuracy and inference time.
>
> References:
>
> [R1] Ze Liu, Yutong Lin, Yue Cao, Han Hu, Yixuan Wei, Zheng Zhang, Stephen Lin, and Baining Guo. Swin transformer: Hierarchical vision transformer using shifted windows. In ICCV, 2021.
>
> [R2] Sitong Wu, Tianyi Wu, Haoru Tan, and Guodong Guo. Pale transformer: A general vision transformer backbone with pale-shaped attention. In AAAI, 2022.
>
> [R3] Ashish Vaswani, Noam Shazeer, Niki Parmar, Jakob Uszkoreit, Llion Jones, Aidan N Gomez, Łukasz Kaiser, and Illia Polosukhin. Attention is all you need. In NeurIPS, 2017.
>
> [R4] Xiaoyi Dong, Jianmin Bao, Dongdong Chen, Weiming Zhang, Nenghai Yu, Lu Yuan, Dong Chen, and Baining Guo. Cswin transformer: A general vision transformer backbone with cross-shaped windows. In CVPR, 2022.
>
> [R5] Bobby He and Thomas Hofmann. Simplifying transformer blocks. In arXiv, 2023.
>
> [R6] Zhuofan Xia, Xuran Pan, Shiji Song, Li Erran Li, and Gao Huang. Vision transformer with deformable attention. In CVPR, 2022.
>
> Q2: The convolution, layer normalization, and attention may result in multiple tensor reshaping operations.
>
> A2: Yes, the convolution operation, layer normalization, and attention operation may lead to multiple tensor reshaping operations, which is why the model complexity of current vision adapter-based methods is always higher than that of the baseline models. However, these operations are not only essential but also inherently present in the existing vision adapter block, which also serves as the main motivation for our work. Therefore, under the current framework, we have made enhancements for both layer normalization and attention to improving the model's memory efficiency, including sharing layer normalization across multiple modules and substituting standard self-attention with cross-shaped self-attention.
>
> Q3: A thorough analysis of memory costs associated with each operation in META and previous approaches.
>
> A3: In Table 4 of the main paper, we provide experimental results for the effectiveness of each component of META under the evaluation of the model's accuracy, complexity, memory, and inference efficiency.  From this table, we can observe that the memory costs associated with META predominantly arise from the Attn branch, which contributes an additional 7.5 GB of memory consumption on ViT-B. In contrast, the incorporation of the FFN branch and the Conv branch does not result in a significant increase in memory usage. The implementation of the cascade strategy necessitates the pre-storage of features across different head layers, leading to an additional memory increase of 0.6 GB on ViT-B.
>
> In Table S3 of the supplementary materials, we compare the proposed META with previous efficient attention methods, including Window Attention[R1], Pale Attention[R2], Dense Attention[R3], CSWindow[R4], SimpleAttention[R5], and Deformable Attention[R6]. From the obtained results presented in the final column of this table, it is evident that our model maintains the lowest memory costs among the compared approaches.

---

> ### Author Response · Authors · 2024-11-18
> **Response to Reviewer oVw4 (part 2)**
>
> Q4: An additional ablation study without the Attn Branch.
>
> A4: Thanks for this suggestion. During the experimental design phase, we have considered experimental settings that involved using only the FFN branch. In our revised Table 4, we included the ablation results on instance segmentation, object detection and semantic segmentation tasks under this setting. It is observed that while utilizing only the FFN branch results in minimal model complexity and memory costs, there is a significant accuracy decline.
>
> Q5: Compare META with other methods on V100.
>
> A5: In our submission, the reported results are measured by A100 GPUs with per-GPU batch size 2. To avoid any ambiguity, we have added a description of the GPU settings for the inference phase in Section 4.1 of the revised version. The details are as: “The reported inference results are measured by A100 GPUs with per-GPU batch size 2.”
>
> Following your suggestion, we conducted an extra inference on V100 GPUs with 32 GB and a per-GPU batch size of 1. We chose Cascade Mask R-CNN for instance segmentation and object detection as the baseline, ViT-B is used as the backbone. The inference results are presented below.
>
> | Methods   | MC   | FPS on A100   | FPS on V100    |
> |---   |---   |---   |---   |
> | ViT-B 	| NA 	| 13.0 	| 3.5 	|
> | ViT-Adapter-B 	| 15.2 	| 8.6 	| 2.4 	|
> | LOSA-B 	| 13.0 	| 9.0 	| 2.6 	|
> | META-B 	| 8.1 	| 11.9 	| 3.2 	|
>
> We can observe that the inference speed is somewhat slower on V100 GPUs. Fortunately, compared with other methods, our method still demonstrates superior performance.
>
> Q6: Details regarding the experimental setup In Table S3.
>
> A6: Following the same settings as in [R7], the attention mechanism is utilized as the ViT-adapter layer. Therefore, during the experimental comparisons, we replace the attention mechanism in the ViT-adapter model with other attention mechanisms to ensure a fair comparison. We have included the detailed settings in Section S5 of the supplementary materials.
>
> Other minor comments:
>
> Q1: About the spatial prior module.
>
> A1: Yes, the used spatial prior module follows the same as the one in [R7]. In Section 3.1 of our revision, we have highlighted this point.
>
> Q2: DeiT in Line 96.
>
> A2: Thanks for this suggestion. We have incorporated DeiT into this section.
>
> Q3: About the ambiguously “which” in Line 182.
>
> A3: Thank you for pointing this out. To eliminate any ambiguity, we have revised the sentence as: "The MEA block is designed to facilitate the interaction between the features extracted from the ViT backbone and the spatial prior module. Our block consists of the attention (i.e., Atte) branch, the feed-forward network (i.e., FFN) branch, and the lightweight convolutional (i.e., Conv) branch."
>
> Q4: About the abbreviation of “Concat” and ” Attn”.
>
> A4: Thanks for this suggestion. Based on your suggestion, we have modified "Cat" to "Concat" in Equations 1 and 3, and changed "Atte" to "Attn."
>
> Q5: Line 223: Should "respectively" be replaced with "sequentially"?
>
> A5: We agree that "sequentially" is a more appropriate choice and have implemented the suggested fix accordingly.
>
> Q6: The use of "common" may introduce ambiguity on Line 166 of Table S2.
>
> A6: Thank you for pointing this out. In this context, "common" refers to the classic normalization method, which is non-shared. To avoid ambiguity, we have revised this to "Non-shared normalization," thereby contrasting it with our proposed shared normalization. This modification enhances the clarity of the table's content.

---

> ### Comment · Reviewer_oVw4 · 2024-11-20
> **Official Comment by Reviewer oVw4**
>
> Thank you for your detailed responses, which have partially addressed my concern (Q3-Q6).
> However, some concerns remain:
> > A1: Thanks for your suggestion... model's inference time.
>
> Table 4 presents only the overall fps when ablating different components. I am interested in understanding how many seconds each component consumes during a single inference of your model. For instance, consider providing information similar to the following:
>
> |component  | Inference Time (s) | Memory (GB) |
> |:---------|:--------------:|------:|
> | component 1 |      xx      | xx |
> | component 2 |      xx      | xx |
> | total  |      xx s      | xx |
>
> > c) The implementation of the cascaded strategy can further enhance the model's inference speed, as this manner allows for the parallel processing of different features
>
> Based on Figure 2 and Section 3.3, the cascaded strategy seems to process each head sequentially rather than in parallel.
>
> > A2: Yes, the convolution operation, layer normalization, and attention operation may ... with cross-shaped self-attention.
>
> The MEA block in your method mixes convolutions, layer normalization, and attention, necessitating tensor reshaping operations.
> Compared to the existing vision adapter block, how many reshaping operations does your method reduce? Could you provide a detailed comparison of the number of reshaping operations required in one MEA block versus those in other adapter blocks?

---

> ### Author Response · Authors · 2024-11-21
> **Response to Reviewer oVw4 (part 3)**
>
> A1: How many seconds each component consumes during a single inference of your model.
>
> Q1: Thanks for this valuable suggestion. Following your suggestion, we conducted a comprehensive evaluation of the inference time and memory costs for each component during the inference process. META-B with Mask R-CNN under the 3$\times$ training schedule for instance segmentation and object detection is used as the baseline. The results are presented in the following table, where "Others" denotes the cascade strategy employed in our method, along with additional essential operations required for the execution of the ViT adapter, including convolutional layers, feature separation, feature concatenation, and feature addition operations. The rationale behind this representation style is that the cascade strategy cannot be executed independently; it necessitates integration with operations involving other features. From this table, we can observe that the time and memory costs during the inference process of our method primarily stems from the Attn. branch.
>
> |    Component    |    Inference Time (s)    |    Memory (GB)    |
> |--------|--------|--------|
> |    Attn Branch.    |    0.06251    |    6.07    |
> |    FFN Branch.    |    0.00377    |    0.04    |
> |    Conv Branch.    |    0.00416    |    0.03    |
> |    Others    |    0.01965    |    2.00    |
> |    Total    |    0.09009    |    8.14    |
>
> A2: The cascaded strategy seems to process each head sequentially rather than in parallel.
>
> Q2: Sorry for the confusion. Yes, the cascade strategy processes each head sequentially. However, it can still facilitate the parallel processing of ViT models by allowing certain operations to be executed concurrently. While the strategy is organized in a cascade manner, each stage can independently and simultaneously process different heads, thereby reducing overall computation time. This strategy effectively utilizes GPU computational resources by overlapping the execution of different stages, resulting in enhanced efficiency in inference, despite the serial nature of the cascaded architecture.
>
> A3: How many reshaping operations does your method reduce? Could you provide a detailed comparison of the number of reshaping operations required in one MEA block versus those in other adapter blocks?
>
> Q3: Thanks for your insightful question. In our MEA block, we employ the cross-shaped attention mechanism aimed at minimizing reshaping operations. Compared to the attention mechanisms employed in other adapter blocks, our approach reduces the demand for reshaping by one instance per execution of a single-head attention. This reduction not only mitigates computational complexity but also decreases memory overhead, thereby enhancing the overall efficiency of the model.
>
> To elucidate this, consider an input tensor with dimensions H $\times$ W $\times$ C, where H denotes height, W denotes width, and C denotes the channel size. In a conventional attention mechanism, the process typically involves three steps, which entail two reshaping operations:
>
> 1) The first reshaping: reshaping the input tensor into a matrix of dimensions (H $\times$ W) $\times$ C.
>
> 2) Computing the attention matrix, generally of size (H$\times$ W) $\times$ (H $\times$ W).
>
> 3) The second reshaping: performing a linear transformation using the attention matrix, which necessitates reshaping back to the dimensions of (H $\times$ W) $\times$ C.
>
> In contrast, the cross-shaped self-attention mechanism requires only a single reshaping operation. Our method operates directly on the original tensor dimensions of H $\times$ W $\times$ C, allowing the resulting attention matrix to be reshaped solely back to H $\times$ W $\times$ C, thereby necessitating only one reshaping step.

---

> > ### Comment · Reviewer_oVw4 · 2024-11-23
> > **Official Comment by Reviewer oVw4 （part 3)**
> >
> > Thanks very much for your response, I am still confused and would like to discuss more details.
> >
> > > Q1:
> >
> > I appreciate the comprehensive evaluation of the inference time and memory costs associated with the cascaded strategy. It will be more clear to provide an evaluation of runtime without cascaded strategy.
> >
> > > Q2: This strategy effectively utilizes GPU computational resources by overlapping the execution of different stages
> >
> > Could your provide some pseudocode for this? During the cascaded strategy, the next (h+1)-th head depends on the output from the h-th head. I wonder how to overlap the execution.
> >
> > > Q3: the cross-shaped self-attention mechanism requires only a single reshaping operation. Our method operates directly on the original tensor dimensions of H
> >  W C, allowing the resulting attention matrix to be reshaped solely back to H W C, thereby necessitating only one reshaping step.
> >
> > Could you provide some pseudocode or source code for the cross-shaped self-attention mechanism and your attention branch. Based on my understanding, the cross-shaped self-attention mechanism still involves the following steps:
> > 1. Reshape the input into stripe of H/s W/s s s c [where s is the stripe size].
> > 2. Compute the single head attention matrix, generating an output of H/s W/s s s s s.
> > 3. Reshape back to H W C.
> >
> > This entire process appears to resemble that of CSwin [1]. Do I miss some details?
> >
> > [1] CSWin Transformer: A General Vision Transformer Backbone with Cross-Shaped Windows

---

> > > ### Author Response · Authors · 2024-11-24
> > > **Response to Reviewer oVw4 (part 4)**
> > >
> > > Thanks for your timely feedback. Below is our response to your new comments.
> > >
> > > Q1: Provide an evaluation of runtime without the cascaded strategy.
> > >
> > > A1: To improve the clarity of the runtime evaluation and in alignment with your suggestions, we performed inference experiments that excluded the use of the cascaded strategy. The results obtained are presented in the table below.
> > >
> > > |   Component   |   Inference Time (s)   |
> > > |------|------|
> > > |   Attn Branch.   |   0.08537   |
> > > |    FFN Branch.    |    0.00431    |
> > > |    Conv Branch.    |    0.00495    |
> > > |    Total    |    0.09463    |
> > >
> > >
> > >
> > > Q2: Pseudocode and explain how to overlap the execution.
> > >
> > > A2: Actually, your understanding is correct. During the cascaded strategy, the next $(h+1)$-th head depends on the output from the $(h)$-th head. By "overlapping the execution of different stages," we mean that the cascaded strategy allows each cascading layer to compute independently, and the computation results of each layer can be released in a timely manner when no longer needed, avoiding the retention of all intermediate results in memory. This on-demand computation and release approach allows subsequent layers or stages to compute without waiting for all previous layers or stages to finish, thereby reducing memory usage, allowing for more efficient utilization of hardware resources, and improving inference efficiency. Below, we provide the pseudocode for the MEA Block to facilitate a more detailed understanding of the computational process of our proposed method.
> > >
> > > ```python
> > > ## Function MEA_Block(F_sp, F_vit)
> > > 1. **Input**:
> > >    - F_sp: Features from SPM
> > >    - F_vit: Features from ViT
> > >
> > > 2. **Shared Layer Normalization**:
> > >    - F_sp_ln = LayerNorm(F_sp)
> > >    - F_vit_ln = LayerNorm(F_vit)
> > >
> > > 3. **Attention Branch**:
> > >    - A_H = Cross_Shaped_Self_Attention_Horizontal(F_sp_ln, F_vit_ln)
> > >    - A_V = Cross_Shaped_Self_Attention_Vertical(F_sp_ln, F_vit_ln)
> > >    - A = Concat(A_H, A_V)
> > >
> > > 4. **Feed-Forward Network (FFN) Branch**:
> > >    - F_temp = Concat(F_sp, F_vit)
> > >    - F_temp = Conv3x3(F_temp)
> > >    - F_temp_ln = LayerNorm(F_temp)
> > >    - F_ffn = MLP(F_temp_ln)p
> > >
> > > 5. **Convolutional Branch**:
> > >    - C = Concat(F_sp, F_vit)
> > >    - C = DC(C)  # Depth-wise Convolution
> > >    - C = GLU(C)
> > >    - C = DC(C)
> > >    - C = GLU(C)
> > >
> > > 6. **Output**:
> > >    - F_out = Conv3x3(Concat(A, F_ffn, C, F_sp, F_vit))
> > >
> > > 7. **Return**:
> > >    - Return F_out

---

> > > ### Author Response · Authors · 2024-11-24
> > > **Response to Reviewer oVw4 (part 5)**
> > >
> > > Q3: Provide some pseudocode or source code for the cross-shaped self-attention mechanism and your attention branch.
> > >
> > > A3: As stated in lines 197-200 of the main paper, the Attn. Branch in our work is consistent with the cross-shaped self-attention. The pseudocode for this operation is provided below.
> > >
> > > Yes, your understanding of the entire process is correct. The cross-shaped self-attention mechanism presents a notable reduction in reshaping operations when compared to conventional attention mechanisms, primarily attributed to its implementation of cross-shaped window attention. This approach emphasizes localized window calculations, facilitating attention computation within designated regions of the feature map, thereby circumventing the necessity to reshape the entire input for global attention. Furthermore, the cross-shaped window architecture allows for the sharing of information across multiple windows without the frequent reshaping typically required, thereby streamlining the computational process. Consequently, this design yields enhanced efficiency in attention operations.
> > >
> > > If you have any further comments or suggestions, please feel free to let us know. Thanks.
> > >
> > > ```python
> > > function cross_shaped_window_attention(x, num_heads, window_size):
> > >     # x: given feature
> > >     # num_heads: head number
> > >     # window_size: window size
> > >
> > >     # Get dimensions
> > >     (batch_size, seq_length, d_model) = shape(x)
> > >
> > >     # Split into multiple heads
> > >     Q, K, V = split_heads(x, num_heads)
> > >
> > >     # Initialize attention output
> > >     attention_output = zeros(batch_size, seq_length, d_model)
> > >
> > >     # Initialize previous head's output for cascaded attention
> > >     previous_Q = zeros(batch_size, seq_length, d_model)
> > >     previous_K = zeros(batch_size, seq_length, d_model)
> > >     previous_V = zeros(batch_size, seq_length, d_model)
> > >
> > >     # Calculate attention for each head
> > >     for head in range(num_heads):
> > >         for position in range(seq_length):
> > >             # Get cross-shaped window indices
> > >             window_indices = get_cross_shaped_window_indices(position, window_size)
> > >
> > >             # Gather Q, K, V for the current window
> > >             if head == 0:
> > >                 Q_window = gather(Q[head], window_indices)
> > >                 K_window = gather(K[head], window_indices)
> > >                 V_window = gather(V[head], window_indices)
> > >             else:
> > >                 Q_window = gather(Q[head], window_indices) + previous_Q
> > >                 K_window = gather(K[head], window_indices) + previous_K
> > >                 V_window = gather(V[head], window_indices) + previous_V
> > >
> > >             # Calculate attention scores
> > >             attention_scores = softmax(Q_window * K_window^T / sqrt(d_k))
> > >
> > >             # Compute the attention output for the current position
> > >             attention_output[position] = attention_scores * V_window
> > >
> > >         # Update previous head's output for the next head
> > >         previous_Q = gather(Q[head], window_indices)
> > >         previous_K = gather(K[head], window_indices)
> > >         previous_V = gather(V[head], window_indices)
> > >
> > >     # Final linear transformation
> > >     attention_output = linear_transform(attention_output)
> > >     return attention_output
> > >
> > > function feed_forward_network(x):
> > >     # Feed Forward Network
> > >     x = ReLU(linear(x))
> > >     x = linear(x)
> > >     return x

---

> > > > ### Comment · Reviewer_oVw4 · 2024-11-25
> > > > **Official Comment by Reviewer oVw4 （part 5)**
> > > >
> > > > Thanks again for your comprehensive response. I would like to delve into some further details.
> > > > > This on-demand computation and release approach allows subsequent layers or stages to compute without waiting for all previous layers or stages to finish
> > > >
> > > > If the next h+1-th head depends on the output from the h-th head, the cascading layer must wait for the previous layer to finish in order to obtain the output from the h-th head and compute h+1-th heads.
> > > >
> > > > Compare the pseudocode of cross_shaped_window_attention with that of CSWin, we note the following differences:
> > > > 1. The stripe size in your method is actually 1.
> > > > 2. CSWin reshape the whole feature map into H/s W/s s s c first as follows:
> > > > ```
> > > > def img2windows(img, H_sp, W_sp):
> > > >     B, C, H, W = img.shape
> > > >     img_reshape = img.view(B, C, H // H_sp, H_sp, W // W_sp, W_sp)
> > > >     img_perm = img_reshape.permute(0, 2, 4, 3, 5, 1).contiguous().reshape(-1, H_sp* W_sp, C)
> > > >     return img_perm
> > > > ```
> > > > while you process each small strip sequentially in a loop. It is like using computation to exchange memory?

---

> > > > > ### Author Response · Authors · 2024-11-25
> > > > > **Response to Reviewer oVw4 (part 7)**
> > > > >
> > > > > Thank you for your prompt feedback and your valuable time.
> > > > >
> > > > > Q1: If the next h+1-th head depends on the output from the h-th head, the cascading layer must wait for the previous layer to finish in order to obtain the output from the h-th head and compute h+1-th heads.
> > > > >
> > > > > A1: Yes, that is indeed the case, but subsequent layers or stages can compute without waiting for **ALL** previous layers or stages to finish. For example, when the model computes the result of the $5$-th head and passes it to the $6$-th head, we can release the computation results of the $5$-th head before. In the PyTorch framework, this mechanism optimizes memory allocation and release by using a technique called "memory pooling."
> > > > >
> > > > > Q2: The stripe size in your method is actually 1.
> > > > >
> > > > > A2: We elegantly argue that this is not true. In our paper, we adopt the same stripe size (also named as "stripe width") as the cross-shaped self-attention, which is set to 1, 2, 7, and 7 for the four stages by default. Here is an illustrative example of how it might be set in code, where we take a stripe size of $7 \times 7$ as an example:
> > > > >
> > > > > ```python
> > > > > class CSWinTransformer:
> > > > >     def __init__(self, stripe_width=(7, 7), ...):
> > > > >         self.stripe_width = stripe_width
> > > > >         ...
> > > > >
> > > > >     def cross_shaped_window_attention(self, x):
> > > > >         # Use self.stripe_width to determine the size of the attention windows
> > > > >         ...
> > > > > ```
> > > > > Q3: It is like using computation to exchange memory?
> > > > >
> > > > > A3: In terms of computation, if the number of heads and the number of stripe widths are consistent with those of the cross-shaped self-attention, then our computation is the same as that of the cross-shaped self-attention; the difference lies in the interaction method for different heads, where we adopt a cascaded strategy.

---

> > > > > > ### Comment · Reviewer_oVw4 · 2024-11-25
> > > > > > **Official Comment by Reviewer oVw4 （part 7)**
> > > > > >
> > > > > > Thank you for your response and your valuable time, I believe our discussions will lead to a deeper understanding of your paper.
> > > > > >
> > > > > > > Yes, that is indeed the case, but subsequent layers or stages can compute without waiting for ALL previous layers or stages to finish. [Response to Reviewer oVw4 (part 7) ]
> > > > > >
> > > > > > >  While the strategy is organized in a cascade manner, each stage can independently and simultaneously process different heads, thereby reducing overall computation time. This strategy effectively utilizes GPU computational resources by overlapping the execution of different stages, resulting in enhanced efficiency in inference, despite the serial nature of the cascaded architecture.. [Response to Reviewer oVw4 (part 3)]
> > > > > >
> > > > > > From our discussions util part 7, I understand that executions are still carried out sequentially, with memory being released after each execution, just like the standard forward function in PyTorch. Different heads at different stage is not processed concurrently, or in an overlapping manner. Before processing the 6th head in the 2nd stage, you process the heads in 1st stage and release the corresponding memory. You can not process 6th head in the 2nd stage and 6th head in the 1st stage concurrently.
> > > > > >
> > > > > >
> > > > > > > In your pseudocode for cross_shaped_window_attention in Response to Reviewer oVw4 (part 5):
> > > > > > ```
> > > > > >  (batch_size, seq_length, d_model) = shape(x)
> > > > > >  for position in range(seq_length):
> > > > > > ```
> > > > > >
> > > > > > It seems you use a stripe size of 1. When the stripe size is 2, do you still use this loop to process each stripe, rather than first converting the image to stripes by img2windows, similar to the approach taken in CSWin?

---

> > > > > > > ### Author Response · Authors · 2024-11-25
> > > > > > > **Response to Reviewer oVw4 (part 8)**
> > > > > > >
> > > > > > > Thanks for your timely comments. With your help, we have indeed gained a deeper understanding of this work and our ideas have become clearer.
> > > > > > >
> > > > > > > Q1. You can not process 6th head in the 2nd stage and 6th head in the 1st stage concurrently.
> > > > > > >
> > > > > > > A1: Yes, you are correct. I apologize for any misunderstanding that may have arisen from my inappropriate phrasing in the response to part 3. What we intended to convey is that the cascaded strategy within the Attn. branch processes each head sequentially, and this on-demand computation and release manner benefits the overall parallel computation of the model by freeing up more memory during the process.
> > > > > > >
> > > > > > > Q2. When the stripe size is 2, do you still use this loop to process each stripe, rather than first converting the image to stripes by img2windows, similar to the approach taken in CSWin?
> > > > > > >
> > > > > > > A2: When the stripe size is 2, we will not use this loop. We adopt the indexing manner to obtain the relevant features, rather than reshaping them into a specific shape by img2windows. This helps to reduce unnecessary reshaping operations.

---

> > > > > > > > ### Comment · Reviewer_oVw4 · 2024-11-25
> > > > > > > > **Official Comment by Reviewer oVw4 （part 8)**
> > > > > > > >
> > > > > > > > Thank you for your insightful discussions. I have addressed my concerns and I believe that the paper should be accepted. I have raised my score accordingly.
> > > > > > > >
> > > > > > > > Meanwhile, could you please include the pseudocode or source code for the cross-shaped self-attention mechanism when the stripe size is set to 2? This is necessary as it differs from the pseudocode you provided, which is applicable to a stripe size of 1.

---

> > > > > > > > > ### Author Response · Authors · 2024-11-26
> > > > > > > > > **Thanks for your reply**
> > > > > > > > >
> > > > > > > > > Thanks for your valuable time, positive comments, and constructive suggestions, as well as the efforts you put into this review process. We will include pseudocode for when the stripe size is set to 2 in the supplementary materials to help readers better understand our method.

---

> > > ### Author Response · Authors · 2024-11-25
> > > **Response to Reviewer oVw4 (part 6)**
> > >
> > > We would like to thank you again for your valuable time, positive comments, and constructive suggestions. As the discussion period is approaching to an end, if you have any further comments or suggestions, please feel free to let us know. Thank you.

---

### Official Review · Reviewer_ZAUt · 2024-10-28

**Soundness:** 3
**Presentation:** 3
**Contribution:** 3
**Rating:** 6
**Confidence:** 4

**Summary:**

This paper explores the limitations of Vision Transformer (ViT) adapters in dense prediction tasks, particularly focusing on the issues of memory inefficiency and slow inference speed caused by frequent reshaping operations and normalization steps. The paper proposes a novel ViT adapter named META, which introduces a memory-efficient adapter block that enables the sharing of normalization layers between the self-attention layer and the feed-forward layer. Furthermore, a lightweight convolutional branch is added to enhance the adapter block. Ultimately, this design achieves a reduction in memory access overhead.

**Strengths:**

This paper presents a simple and fast ViT adapter named META, which addresses the critical yet underexplored issue of memory inefficiency. The quality of this paper is supported by theoretical foundations and empirical validations across various tasks and datasets, demonstrating that META outperforms state-of-the-art models in terms of accuracy and memory usage. The paper is structured clearly, with detailed architectural descriptions and clear explanations of the proposed motivation.

**Weaknesses:**

In the Atte Branch discussed in this paper, the adoption of the cross-shaped self-attention (CSA) mechanism is a pivotal factor in effectively reducing the frequent reshaping operations of the model. However, the current analysis lacks an in-depth comparison and discussion between CSA and other efficient attention mechanisms, failing to fully elaborate on why the selection of CSA achieves the current experimental results.

The ablation analysis in this paper are currently limited to the results of instance segmentation on the MS-COCO dataset, whereas your previous experimental work also encompassed the tasks of object detection and semantic segmentation. Therefore, the current ablation analysis regarding the components of the proposed module has certain limitations in terms of generalization. To more comprehensively evaluate the effectiveness and universality of the module components, I recommend conducting corresponding experimental validations for all three tasks of object detection, instance segmentation, and semantic segmentation, thereby ensuring the accuracy and applicability of the conclusions obtained.

In this paper, there is an inconsistency in the presentation, specifically between Formula (1) and part (a) of Figure 2, which do not align accurately. Although you have explained later in the text that the channel concatenation step for Fsp and Fvit is omitted in the formula, this omission may still lead to misunderstandings among readers. To ensure clarity and accuracy of the content, we recommend that the two should correctly correspond to each other.

**Questions:**

In your experimental section, you have conducted in-depth explorations of the three tasks: object detection, instance segmentation, and semantic segmentation. To more intuitively demonstrate the specific improvements brought by your model in handling these tasks, are there any relevant visualization results to support this?

---

> ### Author Response · Authors · 2024-11-18
> **Response to Reviewer ZAUt**
>
> We extend our sincere gratitude for your confirmation of our work as simple and fast, addressing the critical yet underexplored issue of memory inefficiency, demonstrating state-of-the-art performance in terms of accuracy and memory usage, and a clear structure with detailed architectural descriptions and clear explanations. The following is our response with some necessary explanations. If you have any further questions, please feel free to let us know.
>
> Q1: Comparison and discussion between CSA and other efficient attention mechanisms.
>
> A1: In Table S3 of the supplementary materials, we have compared META under CSA with other efficient attention methods, including Window Attention[R1], Pale Attention[R2], Dense Attention[R3], CSWindow[R4], SimpleAttention[R5], and Deformable Attention[R6]. All methods are evaluated using their default configurations and the same settings as the adapter injector and extractor in the ViT-adapter model[R7] to ensure fairness. As stated in Section S5 of the supplementary materials, the experimental results demonstrate that our method can achieve state-of-the-art performance in terms of both accuracy and efficiency.
>
> References:
>
> [R1] Ze Liu, Yutong Lin, Yue Cao, Han Hu, Yixuan Wei, Zheng Zhang, Stephen Lin, and Baining Guo. Swin transformer: Hierarchical vision transformer using shifted windows. In ICCV, 2021.
>
> [R2] Sitong Wu, Tianyi Wu, Haoru Tan, and Guodong Guo. Pale transformer: A general vision transformer backbone with pale-shaped attention. In AAAI, 2022.
>
> [R3] Ashish Vaswani, Noam Shazeer, Niki Parmar, Jakob Uszkoreit, Llion Jones, Aidan N Gomez, Łukasz Kaiser, and Illia Polosukhin. Attention is all you need. In NeurIPS, 2017.
>
> [R4] Xiaoyi Dong, Jianmin Bao, Dongdong Chen, Weiming Zhang, Nenghai Yu, Lu Yuan, Dong Chen, and Baining Guo. Cswin transformer: A general vision transformer backbone with cross-shaped windows. In CVPR, 2022.
>
> [R5] Bobby He and Thomas Hofmann. Simplifying transformer blocks. In arXiv, 2023.
>
> [R6] Zhuofan Xia, Xuran Pan, Shiji Song, Li Erran Li, and Gao Huang. Vision transformer with deformable attention. In CVPR, 2022.
>
> [R7] Zhe Chen, Yuchen Duan, Wenhai Wang, Junjun He, Tong Lu, Jifeng Dai, and Yu Qiao. Vision transformer adapter for dense predictions. In ICLR, 2023.
>
> Q2: More ablation analysis on object detection and semantic segmentation.
>
> A2: Thanks for your suggestion. In Table 4 of our revised submission, we have included the results of the ablation study experiments on object detection and semantic segmentation. Besides, in Section 4.4 “ablation analysis”, we have added an analysis of the experimental settings and results pertaining to this Table.
>
> Q3: Inconsistency presentation between Formula (1) and part (a) of Figure 2.
>
> A3: Thank you for pointing this out. In our revision, we have fixed Formula 1 by adding the channel concatenation part for  F^{sp}  and F^{vit}  to ensure consistency between Figure 2(a) and Formula 1.
>
> Q4: Visualization results.
>
> A4: The visualized result comparisons are provided in Figures S1 and S2 of the supplementary materials. Specifically, a) In Figure S1, we compare the class activation map for instance segmentation before and after the addition of the Conv branch. From this figure, it is evident that after incorporating the Conv branch, our method focuses more on specific object areas (e.g., "the dog" and "the person") rather than the surrounding regions that may extend beyond the objects themselves. This indicates that our method effectively learns local inductive biases following the integration of the Conv branch. b) In Figure S2, we present qualitative results for object detection, instance segmentation, and semantic segmentation. The results demonstrate that, compared to other methods, our method can achieve more accurate object masks that better align with the ground truth boundaries of the objects.

---

> > ### Comment · Reviewer_ZAUt · 2024-11-21
> >
> > Thank you for your reply. I believe this paper should be above the acceptance threshold.

---

> > > ### Author Response · Authors · 2024-11-21
> > > **Many thanks for your confirmation**
> > >
> > > Thank you for your support of our work and for the diligence you showed in the review process.

---

### Official Review · Reviewer_AfWa · 2024-11-03

**Soundness:** 3
**Presentation:** 3
**Contribution:** 2
**Rating:** 6
**Confidence:** 4

**Summary:**

This paper proposes META, an efficient ViT Adapter that enhances ViT in dense prediction tasks. The adapter block MEA provides the local bias required for image tasks to ViT by introducing conv branches, and significantly reduces memory time consumption by minimizing reshape operations on tensors in the adapter. In classic dense prediction tasks such as Object Detection, Instance Segmentation, and Semantic Segmentation, META outperforms previous adapter methods in terms of fewer parameters and lower memory consumption. Ablation experiments were conducted to verify the effectiveness of the three modules in the MEA block and the improvement of the model with the MEA cascade.

**Strengths:**

1. The method proposed in this work is simple but effective, achieving higher performance and efficiency in various classic detection and segmentation frameworks.
2. The paper provides clear and understandable descriptions of the details of each module in the MEA block, with the design purposes of each module being clear and effective.

**Weaknesses:**

1. There is still space on the main text pages, but the implementation parameters of the model are not clarified, such as the number of cascades. Different designs of each size are also not specified.

**Questions:**

1. In Table 2 and Table 3, models of different sizes have the same Memory Consumption (MC). What specific quantity does MC describe, and what measurements lead to this phenomenon?
2. For different sizes of variants of META, are there differences in the implementation details?

---

> ### Author Response · Authors · 2024-11-18
> **Response to Reviewer AfWa**
>
> We would like to express our sincere gratitude for your acknowledgment of our work as both simple and effective, achieving higher performance and efficiency across various frameworks, and providing clear and understandable descriptions of the details of each module in the MEA block. The following is our response with some necessary explanations. If you have any further questions, please feel free to let us know.
>
> Q1: About the number of cascades.
>
> A1: In our work, the number of cascades is set to 16. A detailed description of the specific implementation for the cascaded MEA injector can be found in Section 3.3, and the implementation details of the cascaded MEA extractor are presented in Section 3.4. The details are as follows:
>
> a) In Lines 253-258 of Section 3.3: “Specifically, we first divide the given features into H parts along the channel dimension in a multi-head manner, following the classic self-attention (Vaswani et al., 2017), where H is set to 16 in our work. In the computation process of each head, the output of the h-th head F ̂_sp^(h,i) and F ̂_vit^(h.i) is added into the input features of the next (h+1)-th head F ̂_sp^(h+1,i-1) and F ̂_vit^(h+1,i) to be used in the calculation of subsequent self-attention features, where h = 1, 2, ...,H. The cascade process continues until the feature from the last head is included in the computation.”
>
> b) In line 269 of Section 3.4: “Similarly, the cascaded mechanism is applied following the cascaded MEA injector until the feature from the last head is included in the computation.”
>
> Q2: Different designs of META variants.
>
> A2: For the various sizes of META variants, there are no differences in the implementation details. We maintain the same configurations for the Attention branch, the FFN branch, and the lightweight convolutional branch.
>
> Q3: What specific quantity does MC describe, and what measurements lead to models of different sizes having the same MC?
>
> A3: Sorry for the confusion. MC describes the amount of GPU storage required of the Adapter block to store model parameters, intermediate activations, feature maps, and gradients during both training and inference processes. This metric is closely related to the settings of input image size, batch size, precision, and model architecture. Since we have maintained consistency in these related settings of the Adapter block across different model sizes, this results in models of varying sizes exhibiting the same MC.

---

> > ### Comment · Reviewer_AfWa · 2024-11-20
> > **Response to Authors**
> >
> > Thank you for clarifying the experimental details. I have no further questions, so I'm happy to raise the score to 6.

---

> > > ### Author Response · Authors · 2024-11-21
> > > **Thanks for your reply**
> > >
> > > Thank you for your acknowledgment of our work and for the efforts you put into this review process.

---

### Author Response · Authors · 2024-11-18
**General Response**

We thank all reviewers for their valuable comments and constructive suggestions. We have made the corresponding revisions to both the main paper and the supplementary materials. The main revisions are summarized as follows:
1. Added more implementation details and ablation results.
2. Some necessary minor revisions.

The details of the revisions are referred to the following official comments.

---

### Meta-Review · Area_Chair_HFfm · 2024-12-20

**Metareview:**

The paper introduces a memory-efficient transformer adapter aimed at improving inference speed and memory efficiency by reducing inefficient memory access operations. The method shares layer normalization across layers, uses cross-shaped self-attention, and includes a lightweight convolutional branch. Experimental results show a strong accuracy-efficiency trade-off. Reviewers raised concerns about the novelty of the approach, lack of comparisons with other attention mechanisms, and incomplete experimental evaluation across all tasks. In response, the authors clarified these points, provided additional results, and addressed inconsistencies in the paper. The AC recognizes the paper’s promising contributions to ViT efficiency, and agrees to accept the paper. The paper is recommended to incorporate the rebuttal insights into the final version.

**Additional Comments On Reviewer Discussion:**

Reviewers raised concerns about the novelty of the proposed methods, particularly the lack of comparison with other efficient attention mechanisms, the need for a more comprehensive experimental evaluation covering all three tasks (object detection, instance segmentation, and semantic segmentation), and the unclear relationship between the model's memory efficiency and the proposed techniques. Furthermore, some inconsistencies were noted between the formula and the associated figure in the paper, which may confuse readers. The authors addressed these concerns in their response, providing clarifications and additional experimental results, including a more detailed analysis of the model's efficiency and the contributions of the various components. The reviwers achieved consistent positive scores.

---

### Decision · Program_Chairs · 2025-01-22

Accept (Poster)